# Intrinsic and induced quantum quenches for enhancing qubit-based quantum noise spectroscopy

Yu-Xin Wang [1✉] & Aashish A. Clerk [1]

Quantum sensing protocols that exploit the dephasing of a probe qubit are powerful and ubiquitous methods for interrogating an unknown environment. They have a variety of applications, ranging from noise mitigation in quantum processors, to the study of correlated electron states. Here, we discuss a simple strategy for enhancing these methods, based on the fact that they often give rise to an inadvertent quench of the probed system: there is an effective sudden change in the environmental Hamiltonian at the start of the sensing protocol. These quenches are extremely sensitive to the initial environmental state, and lead to observable changes in the sensor qubit evolution. We show how these new features give access to environmental response properties. This enables methods for direct measurement of bath temperature, and for detecting non-thermal equilibrium states. We also discuss how to deliberately control and modulate this quench physics, which enables reconstruction of the bath spectral function. Extensions to non-Gaussian quantum baths are also discussed, as is the application of our ideas to a range of sensing platforms (e.g., nitrogen-vacancy (NV) centers in diamond, semiconductor quantum dots, and superconducting circuits).

[1] Pritzker School of Molecular Engineering, University of Chicago, 5640 South Ellis Avenue, Chicago, IL 60637, USA. ✉email: yxwang@uchicago.edu

A key technique in quantum sensing is to use a suitably driven sensor qubit to characterize a noisy, dissipative environment. Commonly referred to as quantum noise spectroscopy (QNS)[1], this modality allows one to understand and possibly mitigate sources of decoherence that degrade a quantum processor[2–19]. It also serves as a powerful means to probe a complicated many-body target system via its fluctuation properties (see, e.g.,[20–24]). While many QNS protocols focus on the more specific problem of characterizing classical Gaussian noise[4–12], recent work has explored methods that go beyond these assumptions[13–19,25–30].

One crucial difference between a true quantum environment and a simple classical noise source is that the former is dynamical: its properties can change in response to an external perturbation. At a simple linear response level, this is encoded in the environment's susceptibility functions, or equivalently, asymmetric-in-frequency quantum noise spectral densities[31–35]. The most direct (and perhaps extreme) way to probe these properties is to induce a quantum quench, where the environment experiences a sudden change in its Hamiltonian. Studying the consequences of deliberate quenches has been an extremely useful tool for probing a variety of phenomena in correlated systems[36].

In this work, we show that the basic physics of a quantum quench is relevant to a wide variety of commonly employed QNS schemes and systems; crucially, this is the case even if the protocol does not involve a deliberate quenching of the environment. We show how these quenches (whether intrinsic or deliberate) can be harnessed as a powerful new sensing modality: they reveal environmental response properties in previously unexplored ways. By analyzing standard $T_2$-type qubit-based QNS protocols, we identify generic conditions under which an inadvertent quench of the environment influences the sensor qubit's evolution. Surprisingly, the existence and properties of this quench effect are not simply a function of the initial environmental Hamiltonian, but instead depend on the initial environmental state. The dominant effect of the quench is an unexpected phase shift of the sensor-qubit coherence. For common cases where the environment is either a Gaussian quantum bath or the sensor–environment coupling is weak, we derive a simple, analytical expression connecting this quench phase shift (QPS) to a dissipative susceptibility of the environment (i.e., an effective density of states, DOS). We then use this to address a number of phenomena. In particular, using the extra information provided by the QPS, a standard $T_2$-based QNS protocol can be enhanced to independently characterize both fluctuation and response properties. As we discuss, such information lets us determine the temperature of a thermal equilibrium environment, making only mild assumptions encompassing a wide range of realistic scenarios (including sub-, super-, and Ohmic environments, environments generating $1/f$ noise, etc.). For the paradigmatic case of an environment with an Ohmic spectral density, we show that one can use the QPS (along with standard decoherence measurements) in a simple Hahn-echo protocol to directly extract the environmental temperature (something that cannot be done from decoherence measurements alone). We also show that the quench mechanism is relevant to generic initial bath states beyond equilibrium, and can be used to probe response properties in nonequilibrium systems. We further discuss extensions of this physics in regimes beyond the validity of linear response.

## Results

### Intrinsic quantum quenches in standard $T_2$-type sensing protocols

While our ideas apply to a wide variety of settings, we focus throughout this paper on a standard QNS experiment where the sensor qubit is coupled to an environment via a pure-dephasing interaction. Transforming to the standard toggling frame set by the choice of qubit-control pulses (see e.g.,[1]), as well as the rotating frame with respect to free qubit Hamiltonian $\Omega\hat{\sigma}_z/2 = (\Omega/2)\big(|{\uparrow}\rangle\langle{\uparrow}| - |{\downarrow}\rangle\langle{\downarrow}|\big)$, the qubit-bath Hamiltonian is given by

$$\hat{H}_{\text{tot}} = |{\uparrow}\rangle\langle{\uparrow}| \otimes \hat{H}_{\text{b},\uparrow} + |{\downarrow}\rangle\langle{\downarrow}| \otimes \hat{H}_{\text{b},\downarrow}, \tag{1}$$

where $\hat{H}_{\text{b},\uparrow}$ ($\hat{H}_{\text{b},\downarrow}$) describes the bath Hamiltonian conditioned on qubit being in the state $|{\uparrow}\rangle$ ($|{\downarrow}\rangle$). We set $\hbar = 1$ throughout. As in standard $T_2$-type measurements, the probe qubit is initialized in $|{\downarrow}\rangle$ and is initially unentangled with the bath. The quench physics we describe is crucially sensitive to the initial state of the bath. For illustrative purposes, we first focus on a simple but generic situation where the qubit $|{\downarrow}\rangle$ state lifetime can be viewed as infinite, and the bath has relaxed to a thermal equilibrium state with respect to $\hat{H}_{\text{b},\downarrow}$. The initial density matrix of the qubit-bath system is thus

$$\hat{\rho}_{\text{tot}}(t = 0^-) = |{\downarrow}\rangle\langle{\downarrow}| \otimes \hat{\rho}_{\text{b,i}}, \tag{2}$$

$$\hat{\rho}_{\text{b,i}} = e^{-\hat{H}_{\text{b},\downarrow}/k_{\text{B}}T}/Z_T, \tag{3}$$

where $Z_T$ is a normalization factor and $T$ is the initial bath temperature. We stress that the initial bath state $\hat{\rho}_{\text{b,i}}$ closely depends on the initial qubit state: again assuming the system has reached thermal equilibrium prior to start of the sensing protocol, and if instead the qubit is initialized in $|{\uparrow}\rangle$, then $\hat{\rho}_{\text{b,i}}$ would be a thermal state with respect to $\hat{H}_{\text{b},\uparrow}$.

We consider a standard $T_2$-based sensing protocol. At the start of the protocol ($t = 0$), an instantaneous $\pi/2$-pulse is applied to prepare the qubit in an equal superposition state $|+\rangle \equiv \big(|{\uparrow}\rangle + |{\downarrow}\rangle\big)/\sqrt{2}$; the system then evolves under $\hat{H}_{\text{tot}}$ for time $t_f$, while the qubit is subject to a sequence of instantaneous control $\pi$-pulses. At the end of the protocol, one measures qubit Pauli operator $\hat{\sigma}_x$ or $\hat{\sigma}_y$. By repeating the measurements and varying $t_f$, one can obtain the qubit coherence $\langle\hat{\sigma}_-(t_f)\rangle$ as a function of $t_f$.

Surprisingly, in many cases an intrinsic effective bath quench occurs as part of this standard sensing protocol. To see this, we first rewrite $\hat{H}_{\text{tot}}$ as

$$\hat{H}_{\text{tot}} = \frac{1}{2}\hat{\sigma}_z \otimes \hat{\xi} + \hat{\sigma}_0 \otimes \frac{1}{2}(\hat{H}_{\text{b},\uparrow} + \hat{H}_{\text{b},\downarrow}), \tag{4}$$

with $\hat{\xi} \equiv \hat{H}_{\text{b},\uparrow} - \hat{H}_{\text{b},\downarrow}$, $\hat{\sigma}_0 \equiv |{\uparrow}\rangle\langle{\uparrow}| + |{\downarrow}\rangle\langle{\downarrow}|$. This suggests a simple picture for understanding the qubit evolution during the protocol: the qubit dephases due to coupling to the bath noise operator $\hat{\xi}$, while the bath evolves under an effective averaged bath Hamiltonian. Note that these two processes are not independent, as the effective bath Hamiltonian would affect $\hat{\xi}$ during time evolution and hence influence qubit dynamics.

Notably, the averaged bath Hamiltonian in Eq. (4) may or may not commute with the initial bath state, which in our example is determined by $\hat{H}_{\text{b},\downarrow}$. As shown in Fig. 1a, this motivates defining a time-dependent effective bath Hamiltonian $\hat{H}_{\text{b,eff}}(t)$ whose form reflects the change of the qubit state $\hat{\rho}_{\text{qb}}$ at $t = 0$:

$$\hat{H}_{\text{b,eff}}(t) \equiv \text{Tr}_{\text{qb}}[\hat{\rho}_{\text{qb}}(t)\hat{H}_{\text{tot}}(t)], \tag{5}$$

$$= \begin{cases} \hat{H}_{\text{b},\downarrow}, & t \le 0, \\ (\hat{H}_{\text{b},\uparrow} + \hat{H}_{\text{b},\downarrow})/2, & 0 < t < t_f. \end{cases} \tag{6}$$

We see that except for the trivial case $\hat{H}_{\text{b},\uparrow} = \hat{H}_{\text{b},\downarrow}$, the bath Hamiltonian $\hat{H}_{\text{b,eff}}(t)$ exhibits a sudden change (i.e., a quench)

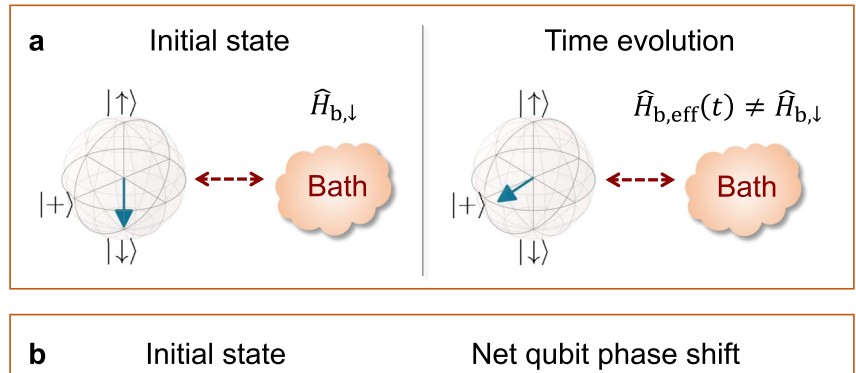

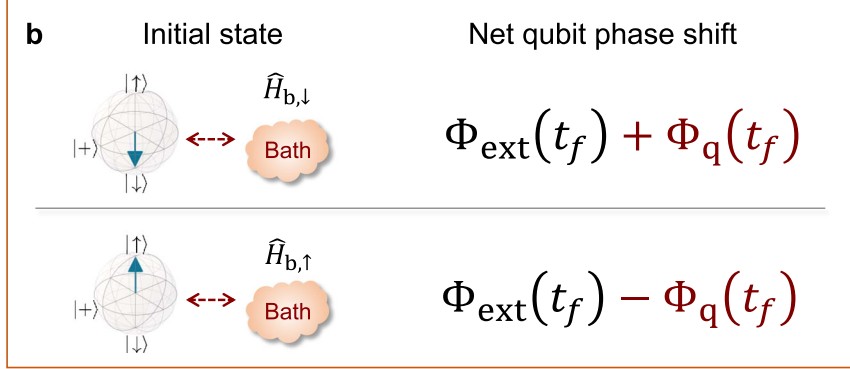

**Fig. 1 Quantum quenches in standard dephasing-based quantum sensing (noise spectroscopy) protocols. a** Schematic illustrating an intrinsic quantum quench arising in standard $T_2$-type experiments, where the effective bath Hamiltonian undergoes a sudden change at the start of the protocol, c.f. Eqs. (5) and (6). **b** The quench manifests itself as an additional quench phase shift (QPS) $\Phi_q(t_f)$ of the sensor qubit. The QPS can be distinguished from a phase $\Phi_{ext}(t_f)$ resulting from an external field: in the simplest case, it is crucially sensitive to the initial qubit state before the start of the sensing protocol.

that is solely due to the sudden change in qubit state at $t = 0$. As we show below in Eqs. (11) and (12), this quench is physically meaningful: it directly determines the evolution of the qubit coherence $\langle \hat{\sigma}_-(t_f) \rangle$, the very quantity that is measured in the protocol.

For our subsequent discussion, it is useful to rewrite $\hat{H}_{b,eff}(t)$ to make the quench more explicit:

$$\hat{H}_{b,eff}(t) = \hat{H}_{b,i} + \eta(t)\hat{V}. \quad (7)$$

Here $\eta(t)$ is an effective quench control function, which encodes the temporal profile of the quench. $\hat{V}$ represents the quench operator, which is defined as

$$\hat{V} \equiv \hat{H}_{b,eff}(t = 0^+) - \hat{H}_{b,eff}(t = 0^-). \quad (8)$$

For the specific example considered here, we have $\hat{H}_{b,i} = \hat{H}_{b,\downarrow}$ and

$$\eta(t) = \Theta(t)\Theta(t_f - t), \quad (9)$$

$$\hat{V} = \hat{\xi}/2, \quad (10)$$

where $\Theta(t)$ is the Heaviside step function. If the total initial state is in thermal equilibrium with respect to $\hat{H}_{tot}$ in Eq. (1), we can again define $\hat{H}_{b,i}$ using Eq. (5) as the bath Hamiltonian contingent on the initial qubit state. The quench operator $\hat{V}$ in this case is sensitive to the initial qubit state: if the qubit was initialized in $|\uparrow\rangle$, then we would have $\hat{H}_{b,i} = \hat{H}_{b,\uparrow}$ and $\hat{V} = -\hat{\xi}/2$. If the total system is initially out of equilibrium, the definitions in Eqs. (7) and (8) can describe quench physics corresponding to a much wider range of nonequilibrium initial bath states, even beyond the specific case in Eqs. (3) and (10). In the more general case, $\hat{H}_{b,i}$ in Eq. (7) is directly controlled by the initial bath state (see "Methods").

We stress that in contrast to conventional quench experiments which require external temporal control of the bath, here the quench is intrinsic to the measurement protocol: it occurs unavoidably simply through the back-action of the qubit on the bath associated with the start of the QNS protocol. While we have discussed a simple example here, the same physics also applies to more general initial bath states and more general quench functions $\eta(t)$. We show explicitly in "Methods" how in general, one can find the form of the effective quench operator $\hat{V}$ from the initial bath state (even if it is a non-thermal state unrelated to $\hat{H}_{b,\uparrow/\downarrow}$). Further, one can also generate more complicated quench functions $\eta(t)$: as we will show, one approach to achieve this is to use a qubit embedded in a multilevel physical system, e.g., a nitrogen-vacancy (NV) center defect in diamond (see discussion following Eq. (33)).

As our approach is more general than the specific example of Eq. (3), in what follows, we will allow $\eta(t)$ to have generic time dependence during the time evolution ($0 < t < t_f$), and we will assume a general $\hat{V}$ (unless specified otherwise).

**General sensor qubit evolution including effective quench.** We now rigorously show how the effective quench physics described in Eqs. (7) and (8) manifests itself in our standard $T_2$-based sensing protocol. We first transform to an appropriate interaction picture, determined by the initial (static) bath Hamiltonian $\hat{H}_{b,i}$, and we again assume the standard toggling frame defined by qubit-control pulses. We thus have time-dependent interaction-picture bath operators $\hat{V}(t)$ and $\hat{\xi}(t)$ whose time dependence is generated by $\hat{H}_{b,i}$. Note that as the initial bath state is stationary in our interaction picture (i.e., $[\hat{\rho}_{b,i}, \hat{H}_{b,i}] = 0$), $\hat{\xi}(t)$ will describe stationary quantum noise: all its correlation functions will respect time-translational invariance.

Working in the above interaction picture, and letting $F(t)$ denote the usual filter function (also known as the switching

function in time domain) that encodes the timing of qubit-control $\pi$-pulses, the time-dependent qubit coherence is given by (see "Methods" for details)

$$\langle \hat{\sigma}_-(t_f) \rangle = \frac{1}{2} \text{Tr}(\hat{U}_\uparrow \hat{\rho}_{b,i} \hat{U}_\downarrow^\dagger), \tag{11}$$

$$\hat{U}_{\uparrow(\downarrow)} = \mathcal{T} \exp\left\{ -i \int_{-\infty}^{+\infty} \left[ \eta(t')\hat{V}(t') \pm \frac{F(t')}{2}\hat{\xi}(t') \right] dt' \right\}. \tag{12}$$

We stress that Eqs. (11) and (12) are valid for a generic form of quench function $\eta(t)$ and operator $\hat{V}(t)$, and not limited to the specific case described by Eqs. (5) and (6). Note crucially that we do not include the quench operator $\hat{V}$ in the definition of our interaction picture. While one could work in this alternate frame, it would obscure the fact that in general, $\hat{V}$ does not commute with the initial bath state. It would also lead to a time-dependent bath noise operator $\hat{\xi}'(t)$ that is nonstationary.

To discuss the sensor qubit evolution, it is convenient to separately parametrize the magnitude and phase of the qubit coherence function in Eqs. (11) and (12):

$$\langle \hat{\sigma}_-(t_f) \rangle = \frac{1}{2} \, e^{-\zeta(t_f)} \, e^{-i\Phi(t_f)}. \tag{13}$$

The effects of the environment are now fully described by the (real, nonnegative) dephasing function $\zeta(t_f)$ (which controls the magnitude of the coherence) and the real bath-induced phase-shift function $\Phi(t_f)$. Standard QNS protocols use information in $\zeta(t_f)$ to probe properties of the environment[1]. As we will now see, due to our effective quench physics, key new features of the environment will also reveal themselves through the unexpected phase shift.

**Quench-induced sensor-qubit phase shift.** The general goal of our QNS protocol is to measure properties of the environment. $T_2$-based QNS protocols typically have a sole focus on the fluctuation properties of the bath, specifically fluctuations of the bath noise operator $\hat{\xi}$. The simplest quantity characterizing these is the symmetrized noise spectral density (NSD) $\bar{S}[\omega]$, given by:

$$\bar{S}[\omega] \equiv \frac{1}{2} \int_{-\infty}^{+\infty} dt e^{i\omega t} \langle \{\hat{\xi}(t), \hat{\xi}(0)\} \rangle \tag{14}$$

The average value here is with respect to the initial bath density matrix $\hat{\rho}_{b,i}$. As discussed in many places (see e.g.,[32]), this quantity is symmetric in frequency, and plays the role of a classical NSD.

Another generic environmental property that is not typically probed in standard $T_2$-based QNS schemes is the dynamical response properties of the bath: how does it change in response to a time-dependent external perturbation? At the simplest linear-response level, this is described by conventional linear response susceptibilities (or equivalently, retarded Green's functions). We will be interested in a particular susceptibility, describing how the average value of the noise operator $\hat{\xi}$ changes in response to a perturbation coupling to the quench operator $\hat{V}$. This is described by the Green-Kubo linear response function (see, e.g.,[37] for a pedagogical introduction)

$$G^R_{\xi V}[\omega] \equiv -i \int_{-\infty}^{+\infty} dt e^{i\omega t} \Theta(t) \langle [\hat{\xi}(t), \hat{V}(0)] \rangle. \tag{15}$$

We stress that in general, this susceptibility is distinct from the NSD $\bar{S}[\omega]$. Hence, being able to measure it would provide new information on the properties of our environment.

We return now to the evolution of our sensor qubit during the QNS protocol. For a generic environment, Eqs. (11)–(13) (which describe the sensor-qubit coherence) can be analyzed

perturbatively in both $\hat{\xi}$ and $\hat{V}$; for the simple example of Eqs. (5) and (6), this amounts to perturbation theory in the qubit-bath coupling. The leading-order contributions to the dephasing and phase-shift functions in Eq. (13) can be succinctly written as (see "Methods")

$$\zeta(t_f) \simeq \int_{-\infty}^{+\infty} \frac{d\omega}{4\pi} |F[\omega]|^2 \bar{S}[\omega], \tag{16}$$

$$\Phi(t_f) = \Phi_q(t_f) \simeq \int_{-\infty}^{+\infty} \frac{d\omega}{2\pi} F^*[\omega]\eta[\omega] G^R_{\xi V}[\omega]. \tag{17}$$

Here we use the notation $z[\omega] \equiv \int_{-\infty}^{+\infty} z(t)e^{i\omega t}dt$ to denote the Fourier transform of a temporal function. We stress that these expressions are valid for a general quench, and not just the specific example described by Eqs. (5) and (6).

Eqs. (16) and (17) are generally valid for generic environments in the weak coupling limit; they also become exact for Gaussian quantum environments (i.e., linear coupling to a bath of independent bosonic modes). This covers many experimentally relevant situations (e.g., environments comprised of phononic or photonic modes[38,39], interacting disordered spin baths[24], $1/f$ charge and flux noise sources[40], etc.). Eq. (16) is a standard textbook expression: at the Gaussian level, the qubit dephasing is controlled by the environmental NSD, weighted by the filter function. In contrast, Eq. (17) is less appreciated: because of the effective quench physics described above, and the dynamical nature of the bath, there is a bath-induced phase shift of the qubit sensor. This phase shift $\Phi_q(t_f)$ depends both on the relevant bath susceptibility, the filter function $F[\omega]$ as well as the quench control function $\eta[\omega]$. As we will see, this phase provides a new route to learning about the environment. Note that the consequences of the quench can also be discussed beyond linear response, as would apply to more general environments and sensor–environment couplings (see "Methods").

We stress that the quench-induced sensor phase shift $\Phi_q(t_f)$ can be accessed in exactly the same type of experiments one would use to sense external DC or AC fields, making use of standard Ramsey, Hahn-echo or more complex dynamical-decoupling sequences[1].

A natural concern is whether this quench phase could be distinguished from more trivial phases resulting from external ambient magnetic fields. In the presence of such fields, the net qubit phase shift in Eq. (13) is now given by

$$\Phi(t_f) = \Phi_{ext}(t_f) + \Phi_q(t_f), \tag{18}$$

$$\Phi_{ext}(t_f) = \int_{-\infty}^{+\infty} \frac{d\omega}{2\pi} F^*[\omega] B_{ext}[\omega], \tag{19}$$

where $B_{ext}(t)$ is the external ambient magnetic field, and the QPS $\Phi_q(t_f)$ is again given by Eq. (17). There is a critical difference between $\Phi_{ext}$ and $\Phi_q$: only the latter is sensitive to the initial state of the qubit (see discussion below Eq. (10)). One can thus easily exploit this feature to distinguish the QPS from other more trivial phase-shift mechanisms, e.g., as depicted in Fig. 1b. Further, we note that this feature lets one distinguish the QPS $\Phi_q(t_f)$ from nontrivial qubit phase shifts due to non-Gaussianity of the noise source (the latter has been studied in e.g., Refs. [14,17]).

We note that related phase shifts were discussed in previous works as an anomalous effect emerging in $T_2$-type QNS protocols in systems with an unusual biased qubit-environment coupling[34,35]. In contrast, as we show the quench-induced phase shifts can in fact arise in a far wider set of systems, including ones with an unbiased coupling that according to previous works, would exhibit no extra phase shift. We again stress that it is the initial bath state (and not the qubit-bath coupling) that plays a

key role in the quench physics. This realization will provide an important new control knob, as one can controllably change the properties of the quench via seemingly subtle changes in the initial bath state. As we discuss, this provides a powerful tool for reconstructing environmental spectral functions.

**Quench phase shift as a means to directly probe environmental density of states.** While the above discussion applies to the most general quench scenario, we will often be interested in cases where the quench Hamiltonian is static in the lab frame once the sensing protocol starts. This corresponds to a quench control function $\eta(t) = \Theta(t)\Theta(t_f - t)$. This is the case for the specific example situation in Eqs. (5) and (6). As we will show in Eqs. (40) and (43), this also encompasses the case of more general forms of quench operator $\hat{V}$ beyond Eq. (10), corresponding to a wide number of $T_2$-based sensing protocols with generic initial bath states.

For the above cases, the QPS can be further recast in a form that only involves the imaginary part of response function $\text{Im}\, G^R_{\xi V}[\omega]$. For spin-echo control pulses satisfying $F[0] \equiv \int_0^{t_f} F(t)\mathrm{d}t = 0$, the expression for the QPS further simplifies (see "Methods")

$$\Phi_q(t_f) = -\int_{-\infty}^{+\infty} \frac{d\omega}{\pi\omega} \, \text{Re}\, F[\omega] \, \text{Im}\, G^R_{\xi V}[\omega]. \quad (20)$$

For a general pulse sequences, $F[\omega]$ above should be replaced by $F[\omega] - F[0]$.

Eq. (20) becomes even more revealing in cases like our example of Eq. (5), where the quench operator $\hat{V}$ is proportional to the noise operator $\hat{\xi}$, $\hat{V} = \beta\hat{\xi}$, where $\beta$ is a real constant. We can thus write:

$$\Phi_q(t_f) = \beta \int_{-\infty}^{+\infty} \frac{d\omega}{\omega} \, \text{Re}\, F[\omega] \, \mathcal{J}[\omega], \quad (21)$$

where we have introduced the environmental spectral function

$$\mathcal{J}[\omega] = -\frac{1}{\pi} \text{Im}\, G^R_{\xi\xi}[\omega], \quad (22)$$

which determines the dissipative response of the environment, and also plays the role of an effective DOS. $\mathcal{J}[\omega]$ also corresponds to the asymmetric part of the (unsymmetrized) quantum noise spectrum[32]. Eq. (21) shows that the QPS provides a direct route to learning about properties of the environmental spectral function, a quantity that plays an important role both in quantum noise theory[38] and in various areas of many-body physics. Note that a related expression was derived in Ref. [34] (though this work did not consider the more general situations analyzed here, see e.g., Eqs. (17) and (20)).

While the importance and utility of $\mathcal{J}[\omega]$ is clear in many contexts, it is useful to provide a simple but ubiquitous example. Consider an environment comprised of independent bosonic modes $b_k$ with $\hat{H}_{b,\downarrow} = \sum_k \Omega_k \hat{b}_k^\dagger \hat{b}_k$ and a noise operator $\hat{\xi}(t) = \sum_k g_k e^{i\Omega_k t} \hat{b}_k^\dagger + \text{H.c.}$. In this case, we have $\mathcal{J}[\omega] = \sum_k g_k^2 \delta(\omega - \Omega_k)$: it is indeed a weighted DOS, with each mode's contribution weighted by its coupling constant. This bosonic bath model can be used to describe a variety of phononic or electromagnetic dephasing environments[38,39]. In this simple bosonic case, $\mathcal{J}[\omega]$ is completely independent of the environmental state. However, we stress that our results in subsequent sections remain valid for interacting baths, and/or baths that are not purely bosonic.

The above example highlights a general fact: to understand whether a large bath NSD $\bar{S}[\omega]$ (as revealed by a standard QNS measurement) is due to a large bath DOS or a large mode-

occupancy (i.e., temperature), one needs to also know the spectral function $\mathcal{J}[\omega]$. As such, the information provided by the QPS provides crucial additional information which complements information provided by the dephasing factor. To see this explicitly, consider the case where the initial bath state $\hat{\rho}_{b,i}$ is in thermal equilibrium at temperature $T$. In this case, the quantum fluctuation-dissipation theorem (FDT) yields[41,42]

$$\bar{S}[\omega] = \pi\mathcal{J}[\omega] \coth\frac{\omega}{2k_B T}. \quad (23)$$

This relation suggests something we will investigate in detail further: if one knows both the noise spectrum and spectral function at a given frequency, one can extract (in a parameter-free manner) the environmental temperature. A standard dephasing-based QNS measurement does not provide sufficient information for such an extraction. Only the extra information provided by the QPS makes this possible. Note that our focus here is on dephasing-type couplings between a sensor qubit and the environment. If one instead had a transverse coupling, then an extended version of $T_1$ relaxometry could also be used in principle to extract $\mathcal{J}[\omega]$, see Supplementary Note 1.

Finally, we point out that the above characterization is useful even in more general situations where the initial bath state is not in thermal equilibrium. In that case, the FDT relation in Eq. (23) can be used to define (at each frequency) an effective temperature $T_{\text{eff}}[\omega]$ (see, e.g.,[32,43,44])

$$\coth\frac{\omega}{2k_B T_{\text{eff}}[\omega]} \equiv \frac{\bar{S}[\omega]}{\pi\mathcal{J}[\omega]}. \quad (24)$$

The fact that this quantity varies as a function of frequency would then be direct evidence of an initial nonequilibrium bath state. This is also the kind of nontrivial information that can be addressed in a standard $T_2$-style QNS protocol using the extra information provided by the QPS.

**Using the quench phase shift to probe low-frequency environmental properties and non-thermal states.** As an example of its utility, we show here how the QPS can be used to extract low-frequency spectral properties of a generic environment (encompassing sub-, super-, and Ohmic cases), going beyond what could be done by studying the dephasing factor alone. This information directly allows one to determine if the bath is in thermal equilibrium, i.e., whether the FDT relation of Eq. (23) is violated. In the case where the bath is in equilibrium, it provides a direct means to extract the environmental temperature. While the estimation protocol we discuss here applies to general quench operators, for concreteness we focus on the specific quench configuration in Eq. (10), where $\hat{V}(t) = \hat{\xi}(t)/2$. The protocol applies essentially the same way for more general situations as long as the lab-frame quench operator is static during the protocol (i.e., Eq. (20) must hold).

Our focus here is on a very generic scenario where both the environmental symmetrized noise spectrum and spectral function exhibit power-law behavior at low-frequency limit:

$$\bar{S}[\omega] \sim S_0 \omega^p \quad (\omega \to 0^+), \quad (25)$$

$$\mathcal{J}[\omega] \sim \frac{A_0}{\pi} \omega^s \quad (\omega \to 0^+). \quad (26)$$

Note this includes the case where these quantities tend to a constant asymptotically as $\omega \to 0^+$. Note also that even if one or both of the exponents $p, s$ are negative, Eqs. (25) and (26) can still describe a physical bath, as long as one also introduces a low-frequency IR cutoff. We thus have four parameters characterizing the low-frequency features of the environment. As shown in Eq. (24), a general, non-thermal environment can always be

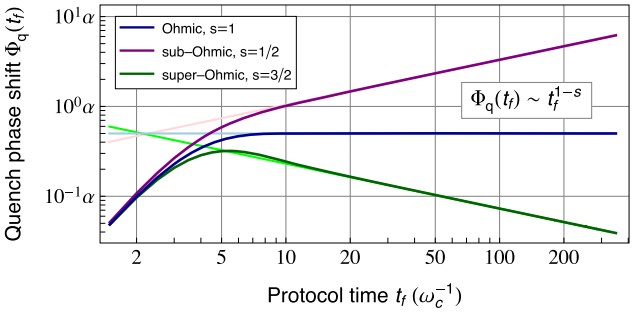

**Fig. 2 Quench phase shift (QPS) $\Phi_q(t_f)$ of a sensor qubit coupled to a Gaussian quantum environment, acquired during a Hahn-echo sequence, as a function of protocol time $t_f$.** We assume the environment has spectral function which behaves as a power law at low frequencies, i.e., $\mathcal{J}[\omega] = (\alpha/\pi)\omega_c(\omega/\omega_c)^s e^{-(\omega/\omega_c)^2}$. Curves correspond to different power laws: Ohmic ($s = 1$, dark blue curve), sub-Ohmic ($s = 1/2$, purple curve), and super-Ohmic ($s = 3/2$, dark green curve). We see that the QPS is extremely sensitive to the spectral function power law $s$. Light-colored lines depict the asymptotic long-time dependence of the QPS $\Phi_q(t_f) \sim t_f^{1-s}$ (see also Eq. (29)), which shows excellent agreement with the exact results in the long-time regime $t_f \gg \omega_c^{-1}$, as expected. Note that for Gaussian, bosonic environments, the QPS is independent of temperature.

characterized by a frequency-dependent effective temperature $T_{\text{eff}}[\omega]$. Using the asymptotic forms given above, we have in the low-frequency limit:

$$T_{\text{eff}}[\omega] \sim \frac{S_0}{2k_B A_0}\omega^{p+1-s}. \qquad (27)$$

If the environment is in thermal equilibrium then $T_{\text{eff}}[\omega]$ will be frequency independent and equal to the bath temperature. We see this requires $p = s - 1$.

Our goal is thus to estimate the power-law exponents $p$, $s$, and overall coefficients $S_0$, $A_0$ from the sensor qubit dynamics. As we now show, this can be achieved by looking at both the phase and magnitude of the qubit coherence in the long-time limit. As long as the asymptotic power-law dependence of bath NSD (response function) does not exhibit too strong a low-frequency divergence, the asymptotic long-time behavior of the dephasing function $\zeta(t_f)$ (QPS $\Phi_q(t_f)$) under any specific spin-echo or dynamical-decoupling pulse becomes independent of details about the cutoff, and is solely determined by the low-frequency asymptotic behavior of NSD (spectral function) in Eqs. (25) and (26). The needed conditions are satisfied by most physical environments (including, e.g., Ohmic baths and baths producing $1/f$ noise).

Using Eqs. (16) and (20) we can rigorously show (see Supplementary Note 2 for a detailed derivation)

$$\zeta(t_f) \sim \mathcal{C}_\zeta S_0 t_f^{1-p} \quad (t_f \to +\infty, -3 < p < 1), \qquad (28)$$

$$\Phi_q(t_f) \sim \mathcal{C}_\Phi A_0 t_f^{1-s} \quad (t_f \to +\infty, -2 < s < 2), \qquad (29)$$

where $\mathcal{C}_\zeta$ and $\mathcal{C}_\Phi$ are nonzero dimensionless coefficients determined by details of the qubit-control pulse. Eqs. (28) and (29) are valid for some of the most common types of physical environments, including Ohmic baths ($p = 0$, $s = 1$) and baths generating $1/f$ noise ($p = -1$, $s = 0$). Comparing against Eq. (27), we see that the combined information in the dephasing function and QPS is exactly what is needed to characterize the effective temperature $T_{\text{eff}}[\omega]$. If this quantity is frequency dependent, the bath is not in a thermal state. Note that for exponents $p$ and $s$ falling out of the range of validity given in Eqs. (28) and (29), the long-time regime of qubit dynamics would also be sensitive to details of the cutoff, but it would still be possible to extract information about the bath NSD (response function) from the

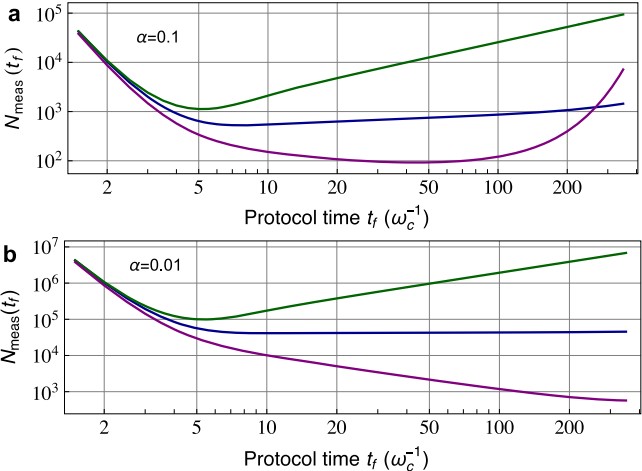

**Fig. 3 Minimum number of measurements $N_{\text{meas}}$ needed to resolve the quench phase shift with a unit signal-to-noise ratio in a Hahn-echo protocol of time $t_f$: $N_{\text{meas}}(t_f) \equiv |\langle \hat{\sigma}_y(t_f)\rangle|^{-2}$.** Because of bath-induced dephasing, it will in general take many repeated measurements to resolve the quench phase shift (QPS). We use the same Gaussian baths (and labeling) as in Fig. 2. Panel (**a**) corresponds to a dimensionless coupling parameter $\alpha = 0.1$, while (**b**) corresponds to $\alpha = 0.01$. All plots correspond to a temperature $k_B T = 0.01\omega_c$.

dephasing function $\zeta(t_f)$ (QPS $\Phi_q(t_f)$) using parametric spectral estimation techniques.

The asymptotic result in Eq. (28) for dephasing is well established[45–47] and has been utilized for QNS in various experimental platforms[4,7,11]. The corresponding result for the QPS in Eq. (29) provides complementary information, on the properties of the spectral function. We stress that to assess whether the bath is in equilibrium, and if so what the temperature is, both these quantities are needed. In Fig. 2, we show the evolution of the QPS for a simple Hahn-echo pulse sequence; curves correspond to Gaussian Ohmic, sub-, and super-Ohmic baths with Gaussian cutoffs, where $s = 1, \frac{1}{2}, \frac{3}{2}$, respectively. As expected, the exact QPS is accurately described by the asymptotic power-law function in the long-time regime. For Hahn echo, the constants appearing in Eqs. (28) and (29) are given by $\mathcal{C}_{\zeta,\text{H}} = \frac{1-2^{p+1}}{\pi}\Gamma(p-1)\sin\frac{p\pi}{2}$ and $\mathcal{C}_{\Phi,\text{H}} = \frac{1-2^s}{\pi}\Gamma(s-1)\cos\frac{s\pi}{2}$, where $\Gamma(\cdot)$ is the gamma function.

We have shown that the long-time properties of the sensor-qubit coherence (both its magnitude and phase) reveal key features of our environment. This sensing modality of course has a natural tension: in the long-time limit, the loss of qubit coherence described by Eq. (16) will make it difficult to resolve the QPS (c.f. Eq. (17)). This is not a fundamental problem, but necessitates sufficient averaging, i.e., repeated evolutions and measurements of the sensor qubit under the chosen pulse protocol. In what follows, we characterize the amount of averaging needed for given environmental parameters.

For convenience, in what follows we express the coefficient $A_0$ in Eq. (26) as $A_0 = \alpha\omega_c^{1-s}$, i.e., the product of a dimensionless parameter $\alpha$ quantifying the qubit-bath coupling strength, and powers of a UV-cutoff frequency scale $\omega_c$ characterizing the regime where Eq. (26) is valid. In the weak coupling limit $\alpha \ll 1$, we can calculate the number of repeated measurements required to achieve a unit signal-to-noise ratio (SNR) for the measurement of the QPS. Focusing only on fundamental projection noise, this is given (as is standard) by the squared inverse norm of the qubit coherence signal[1], $N_{\text{meas}}(t_f) = |\langle \hat{\sigma}_y(t_f)\rangle|^{-2}$. This figure-of-merit is plotted in Fig. 3 for weakly coupled baths with different spectral

functions $\mathcal{J}[\omega]$. Note that in many experimentally relevant situations, the effective environment temperature scale is much lower than the UV energy scale, i.e., $k_B T_{eff} \ll \omega_c$. As a result, measuring the long-time QPS is within reach of state-of-the-art systems realizing QNS.

**Case study: thermometry for low-frequency $1/f$ noise sources.** In this subsection, we focus on low-frequency $1/f$ noise, where the NSD $\bar{S}[\omega] \propto 1/f^a$ ($f < k_B T$, $0 < a < 2$; often $a$ is close to 1). The $1/f$ noise constitutes a dominating dephasing noise source in semiconductor and superconducting qubits[40]. We first show the asymptotic long-time behavior of the QPS $\Phi_q(t_f)$ can now be recast into a simple form, in terms of a few experimentally relevant parameters. We also compute the Hahn-echo signal corresponding to two realistic charge noise models, which can be readily measured using superconducting qubits.

For low-frequency $1/f^a$ noise, the NSD $\bar{S}[\omega]$ satisfies Eq. (25) with a power-law exponent $p = -a$. Assuming the corresponding quantum bath is in thermal equilibrium, we can reformulate the asymptotic results in Eqs. (28) and (29) in terms of three parameters: Hahn-echo dephasing time $T_{2e}$, the noise exponent $a$, and temperature $T$. More specifically, the asymptotic qubit dephasing function $\zeta(t_f)$ in Eq. (28) now reads $\zeta(t_f) \sim (t_f/T_{2e})^{1+a}$, so that we can rewrite the QPS in Eq. (29) as

$$\Phi_q(t_f) \sim \frac{a+1}{2k_B T T_{2e}} \left( \frac{t_f}{T_{2e}} \right)^a \quad (t_f \to +\infty, -1 < a < 3). \quad (30)$$

Thus, to observe the QPS effect it is desirable to have qubits whose Hahn-echo coherence times are smaller than or comparable to the timescale set by temperature, i.e., $T_{2e} \lesssim (k_B T)^{-1}$.

While the asymptotic result in Eq. (30) remains valid for low-frequency charge noise as well as $1/f$ flux noise, we now focus on the former case to estimate the QPS effects in realistic superconducting qubits. In this case we can approximate $T \sim 10^1$ mK, corresponding to a timescale of 1 ns. Experimentally, one can deliberately build superconducting qubits sensitive to charge noise[48] so that QPS effects become measurable.

To obtain a concrete estimate, we adopt two physically motivated microscopic model for charge noise, both consisting of two-level fluctuators (TLFs) coupled to a phonon bath (see[49] for details). Making use of the bath spectral function generated by the models, we numerically compute qubit coherence signal $\langle \hat{\sigma}_y(t_f) \rangle$ under Hahn echo at two different temperatures ($T = 10$ and $50$ mK). The results are plotted against rescaled protocol time $t_f/T_{2e}$ in Fig. 4, where we set qubit–bath coupling such that the qubit coherence time is $T_{2e} = 1$ μs at $T = 10$ mK. We note that the two different models lead to a marked difference in the QPS $\Phi_q(t_f)$, and hence the Hahn-echo signal as one increases temperature.

While the $1/f^a$ noise spectrum has been observed in numerous experiments (see e.g.,[50,51]), we reiterate that the QPS $\Phi_q(t_f)$ offers a direct means to probe the low-frequency bath spectral function $\mathcal{J}[\omega]$, an independent spectral property from the standard NSD $\bar{S}[\omega]$. This is important for understanding decoherence mechanisms in recent designs of superconducting qubits with intrinsic noise protection[52]. On the other hand, despite existing theoretical models that could successfully capture multiple aspects of the $1/f^a$ noise spectra[40], a consensus on the microscopic origin of such noise remains elusive. Having access to the noise temperature (as opposed to the ambient temperature) can also help resolve the open question of origin of $1/f^a$ noise.

**Case study: direct thermometry for an Ohmic bath.** Baths with an Ohmic spectral density $\mathcal{J}[\omega]$ are both extremely well-studied theoretically, and are good descriptions of various dissipative

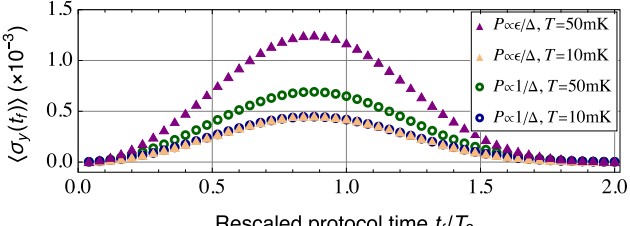

**Fig. 4 Numerically simulated Hahn-echo qubit coherence function $\langle \hat{\sigma}_y(t_f) \rangle$ corresponding to two charge-noise baths.** The bath spectral functions are generated by an ensemble of two-level fluctuators (TLFs) with $\hat{H}_{TLF} = \epsilon \hat{\Sigma}_z + \Delta \hat{\Sigma}_x$, which are coupled to a phonon bath via $\hat{\Sigma}_x$ (see[49] for details). Two TLF distributions $P(\epsilon, \Delta)$ are used in the simulation: $P(\epsilon, \Delta) \propto \epsilon/\Delta$ (triangles), and $P(\epsilon, \Delta) \propto 1/\Delta$ (circles). The qubit–bath coupling strength is chosen such that the qubit coherence time is $T_{2e} = 1$ μs at $T = 10$ mK (orange triangles and blue circles). As we increase bath temperature to $T = 50$ mK, the two models predict qualitative difference in the behavior of the Hahn-echo signal $\langle \hat{\sigma}_y(t_f) \rangle$ (purple triangles and green circles), which in turn encodes dynamics of the quench phase shift (QPS).

environments[53]. Perhaps the best known examples are the voltage and current fluctuations (i.e., Johnson-Nyquist noise[54,55]) of an electromagnetic environment described by an impedance that is frequency independent at low frequencies. Such electromagnetic environments are relevant to many systems, including superconducting qubits[56–59]. In this subsection, we specialize to the case of an environment that is approximately Ohmic at low frequencies, i.e., the low-frequency spectral function $\mathcal{J}[\omega]$ is proportional to frequency. As we now show, this sole assumption allows one to directly extract the environmental temperature via simple measurements that require no curve fitting.

When in thermal equilibrium, an Ohmic environment has a flat NSD at low frequencies, i.e., $\bar{S}[\omega] \sim 2A_0 k_B T$, c.f. Eqs. (25) and (26). A measurement of the low-frequency NSD alone only yields the product of $A_0$ and $T$, and hence does not permit direct thermometry. Luckily, the missing information (i.e., the value of the coupling constant $A_0$) is directly provided by the long-time limit QPS. Defining $\Phi_q(\infty) \equiv \lim_{t_f \to +\infty} \Phi_q(t_f)$, we find from Eq. (29)

$$\Phi_q(\infty) = \frac{\pi}{2} \frac{d\mathcal{J}[\omega]}{d\omega} \bigg|_{\omega=0^+} = \frac{1}{2} A_0. \quad (31)$$

See Supplementary Note 3 for an alternative, intuitive derivation of this expression.

Given this simple result, one can now directly extract the environment temperature. Using the fact that for a Hahn-echo sequence, the $T_2$ decoherence time is given by $T_2 = 2/\bar{S}[0]$[1], we obtain

$$k_B T = \frac{1}{2 T_2 \Phi_q(\infty)}. \quad (32)$$

The upshot is that for a thermal, Ohmic environment, simply measuring the Hahn-echo $T_2$ and the long-time QPS directly yields the environmental temperature. We stress that this does not require any curve fitting, nor further assumptions. Note that our protocol is also applicable if in addition to low-frequency Ohmic noise, we also have large quasistatic noise; see Supplementary Note 4 for details. This is a common scenario in many systems.

While Eq. (32) is exact, it is also useful to understand how long one must wait to achieve the asymptotic long-time limit of the QPS. The answer to this question depends on features in the spectral function away from $\omega = 0$. For convenience, in following discussion we rewrite the spectral function as $\mathcal{J}[\omega] = (\alpha/\pi)\omega\phi(\omega/\omega_c)$, where $\phi(\cdot)$ encodes high-frequency

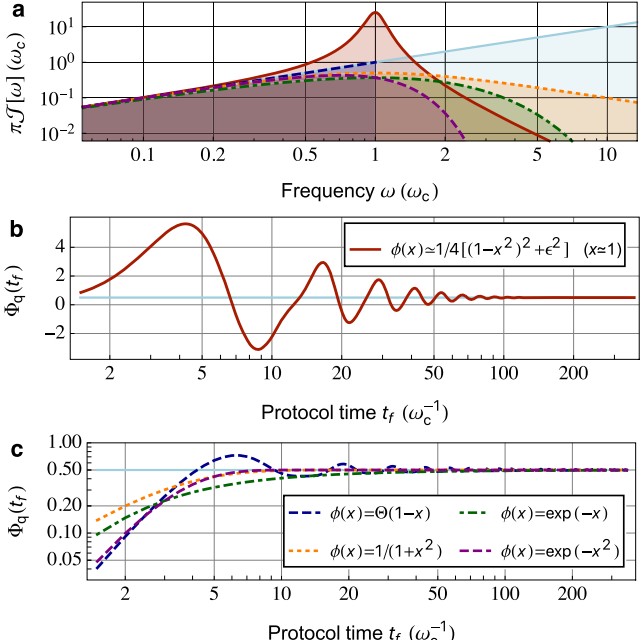

**Fig. 5 Crossover dynamics of the Hahn-echo quench phase shift (QPS) $\Phi_q(t_f)$ for environments that are Ohmic at low frequencies.** For practical applications, the crossover timescale at which the QPS approaches the asymptotic power-law behavior (c.f. Eq. (29)) becomes important. This timescale depends on high-frequency deviations in the bath spectral function $\mathcal{J}[\omega]$ from power law; plotted here are two generic scenarios for this crossover dynamics. As shown in (**a**), we consider bath spectral functions that are asymptotically Ohmic (i.e., proportional to frequency) at low frequencies. **b** Crossover dynamics for QPS with spectral function exhibiting a narrow Lorentzian peak in the range of high frequencies, i.e., $\mathcal{J}[\omega] \sim \alpha\omega_c^3/4\pi[(\omega - \omega_c)^2 + \Gamma_{min}^2]$ for $\omega \simeq \omega_c$ (red solid curve in (**a**)). For narrow peak with linewidth $2\Gamma_{min}/\omega_c = 0.2 < 1$, the crossover timescale is given by $\Gamma_{min}^{-1}$. **c** QPS for spectral functions that are Ohmic at low frequencies with a simple UV cutoff. In this case, the crossover time is given by inverse of the cutoff frequency $\omega_c^{-1}$, and is independent of details about the cutoff. Cutoff functions $\phi(x)$ used: step-function (dashed blue), Lorentzian (dotted orange), exponential (dot-dashed green), and Gaussian cutoff (dashed purple curve). We assume $\alpha = 1$ in all panels. See the main text for specific forms of $\mathcal{J}[\omega]$ used in (**b**) and (**c**).

dependence of the spectral function. As illustrated in Fig. 5, there are two possible scenarios for the crossover dynamics of QPS. First, if the spectral function $\mathcal{J}[\omega]$ exhibits narrow peak(s) in the high-frequency regime, then the crossover timescale is given by $\Gamma_{min}^{-1}$, where $\Gamma_{min} < \omega_c$ is the smallest linewidth of these peaked features. This is shown in Fig. 5b, where the corresponding spectral function exhibits a high-frequency narrow Lorentzian peak with linewidth $2\Gamma_{min} = 2\epsilon\omega_c = 0.2\omega_c$, as encoded by $\phi(x) = (1 + \epsilon^2)^2/[(x - 1)^2 + \epsilon^2][(x + 1)^2 + \epsilon^2]$. Such spectral function can describe e.g., low-frequency photon shot noise generated by a driven damped cavity[29] (see also Supplementary Note 5 for details). The second generic case is where there are no such sharp features at high frequencies, and only a smooth cutoff in $\mathcal{J}[\omega]$ characterized by the UV cutoff frequency $\omega_c$. In this case, the timescale for the QPS to saturate is $1/\omega_c$ and independent of specific details of the form of the cutoff. This is confirmed in Fig. 5c, where QPS crossover dynamics is plotted for step-function cutoff $\phi(x) = \Theta(1 - x)$ (dashed blue), Lorentzian cutoff $\phi(x) = 1/(1 + x^2)$ (dotted orange), exponential cutoff $\phi(x) = e^{-x}$ (dot-dashed green), and Gaussian cutoff $\phi(x) = e^{-x^2}$ (dashed purple curve), respectively.

**Generalized quenches for frequency-space reconstruction of response functions.** In this subsection, we restrict attention to situations where the quench (whether intentional or accidental) yields a quench operator $\hat{V}$ (c.f. Eq.(7)) which commutes with the noise operator $\hat{\xi}$ (Eq. (4)). In the simple and standard case where the quench temporal function $\eta(t)$ is a step function (c.f. Eq. (9)), we showed in Eqs. (25)–(29) that the QPS can be used to extract the low-frequency properties of the environment's spectral function $\mathcal{J}[\omega]$ (i.e., response function). A natural question is to ask whether it is possible to perform a complete reconstruction of $\mathcal{J}[\omega]$ in some finite bandwidth window. This would be then analogous to spectral reconstruction techniques used in conventional QNS measurements to reconstruct $\bar{S}[\omega]$.

It is worth noting that for the specific QPS given by Eq. (21), Ref.[34] has proven a no-go theorem, which prevents systematical reconstructions of spectral function $\mathcal{J}[\omega]$ using the restricted form of quenches in Eqs. (5) and (6). Here we are interested in a more general question: can we utilize quenches with a more complex time-dependence, as encoded in quench function $\eta(t)$, to overcome the limitation set by aforementioned no-go theorem? Indeed, as we show in Supplementary Note 6, the extra tunability in the quench function allows us to use the more general form of QPS in Eq. (17) and reconstruct $\mathcal{J}[\omega]$ in a generic target frequency range.

The protocol we introduce below makes use of a generic structure, where the sensor qubit is controllably embedded in a multilevel system. While this can be realized in many different experimental platforms (e.g., a superconducting transmon qubit, as implemented in Ref.[19]), we focus here on sensor based on a $S = 1$ NV defect in diamond[22]. For this system, we discuss a specific protocol to reconstruct finite-frequency spectral function $\mathcal{J}[\omega]$ by engineering time-dependent quenches. We stress that our strategy can be used to implement generic forms of quench functions $\eta(t)$.

We start by showing how to engineer time-dependent quenches using NV centers in diamond. NV-based qubits are an ideal candidate to implement $T_2$-style QNS: the spin relaxation timescale $T_1$ of NV centers is typically much longer than the dephasing timescale, so that $\hat{S}_z$ is conserved to a great approximation during $T_2$-type protocols. The dominating dephasing typically comes from coupling to environmental magnetic noise (due to surrounding nuclear spins, etc.); alternatively, this makes them a powerful magnetic sensor. We can thus write NV-bath Hamiltonian as

$$\hat{H}_{NV-bath} = \sum_{m_z = 0, \pm 1} |m_z\rangle\langle m_z| \otimes \hat{H}_{b,m_z}. \tag{33}$$

We will consider the NV–bath coupling to correspond to an effective bath-induced magnetic field $\hat{B}$, which then satisfies $\hat{B} = \hat{H}_{b,0} - \hat{H}_{b,-1} = \hat{H}_{b,+1} - \hat{H}_{b,0}$.

The most common and straightforward way to experimentally initialize the NV center is via optical illumination, which prepares it in the $|m_z = 0\rangle$ state[60]. Given this specific initial NV center state, the initial bath Hamiltonian $\hat{H}_{b,i}$ in Eq. (7) should be replaced by $\hat{H}_{b,0}$ (i.e., the bath Hamiltonian conditioned on qubit in $|m_z = 0\rangle$ state). Turning to the sensing protocol, it is relatively easy and straightforward to rapidly produce a superposition state using any two of the three $|m_z\rangle$ states. This provides us then with three different choices for the specific form of the sensor qubit, each corresponding to different effective quench physics. This is summarized in Table 1: the form of the quench physics given by Eqs. (7) and (8) can be controlled or even turned off by choosing the sensor qubit subspace: $\{m_z = 0, m_z = 1\}$, $\{m_z = 0, m_z = -1\}$ or $\{m_z = +1, m_z = -1\}$. This extra knob in NV-based qubits can

**Table 1 Quench operator dependence on subspaces of NV center used to form the sensor qubit, assuming fixed initial NV state $|m_z = 0\rangle$ [a].**

| $|\uparrow\rangle$ | $|\downarrow\rangle$ | Quench operator $\hat{V}$ | Noise operator $\hat{\xi}$ |
|---|---|---|---|
| $|m_z = 0\rangle$ | $|m_z = -1\rangle$ | $-\hat{B}/2$ | $\hat{B}$ |
| $|m_z = +1\rangle$ | $|m_z = 0\rangle$ | $+\hat{B}/2$ | $\hat{B}$ |
| $|m_z = +1\rangle$ | $|m_z = -1\rangle$ | $0$ | $2\hat{B}$ |

[a]The environment is coupled magnetically to the NV spin with $\hat{B} \equiv \hat{H}_{b,0} - \hat{H}_{b,-1} = \hat{H}_{b,+1} - \hat{H}_{b,0}$ (see Eq. (33)).

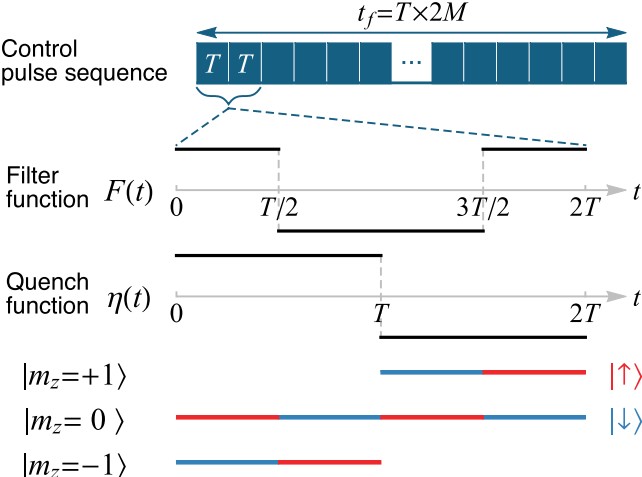

**Fig. 6 Schematic for example NV center control pulse sequence realizing time modulations in both the noise filter function $F(t)$ and quench function $\eta(t)$.** As shown in Eqs. (38) and (39), this control sequence in turn enables reconstruction of spectral function $\mathcal{J}[\omega]$. The control pulses are periodically structured as $2M$ repetitions of a base sequence (period $T = t_f/2M$), so that $F(t)$ and $\eta(t)$ have period $2T = t_f/M$. The noise filter function $F(t)$ encodes timings of the standard dynamical-decoupling $\pi$-pulses within the qubit subspace. In addition, we can further engineer a modulating quench function $\eta(t)$ by periodically switching between qubit subspaces of $\{m_z = 0, m_z = -1\}$ and $\{m_z = 0, m_z = +1\}$ (see also Table 1). The corresponding time-dependent NV levels used as sensor qubit states during the protocol are illustrated in the bottom part of the schematic.

be used to distinguish the quench phase-shift effect from other spurious phases due to the environment[35].

We can now harness this freedom to generate a powerful new kind of quench protocol. The basic idea is to engineer a nontrivial time-dependence of the quench function $\eta(t)$ (c.f. Eq. (7)) by deliberately switching the sensor spin between the different possible qubit subspaces at prescribed times during the protocol. As we show below, the time-dependent $\eta(t)$ generated by this approach can be utilized to generate comb-based filter functions that enable spectral reconstruction of the response function. Figure 6 illustrates a concrete example of control pulses that realize such time-dependent quenches: by periodically switching between the $\{m_z = 0, m_z = -1\}$ and $\{m_z = 0, m_z = +1\}$ qubit subspaces (pulse period $T = t_f/2M$) in addition to applying standard qubit-control $\pi$-pulses at $\ell T/2$ ($\ell = 1, 3, \ldots, 4M - 1$), we effectively realize the more general quench in Eq. (7) with $\hat{H}_{b,i} = \hat{H}_{b,0}$ and $\hat{V} = -\hat{B}/2$, whereas the corresponding quench function $\eta(t) = \sum_{n=0}^{2M-1} (-)^n \Theta(t - nT)\Theta(nT + T - t)$ is shown in Fig. 6.

For the more general control sequence discussed above, the qubit dynamics is again given by Eqs. (11) and (12). The lab-frame noise operator is given by $\hat{\xi} = \hat{B}$; as before, we transform to the toggling frame defined by the standard qubit-control $\pi$-pulses (the control pulses switching between qubit subspaces do not contribute here), with the resulting filter function shown in Fig. 6.

We thus obtain the filter function $F[\omega]$ and quench function $\eta[\omega]$ in frequency space as

$$F[\omega] = -\frac{4}{\omega} e^{\frac{\omega t_f}{2}} \sin \frac{\omega t_f}{2} \frac{\sin^2 \frac{\omega t_f}{8M}}{\cos \frac{\omega t_f}{4M}}, \qquad (34)$$

$$\eta[\omega] = -\frac{2i}{\omega} e^{\frac{i\omega t_f}{2}} \sin \frac{\omega t_f}{2} \tan \frac{\omega t_f}{4M}. \qquad (35)$$

By substituting the above equations into Eq. (17), together they generate a frequency-comb filter function in the large pulse number $M \gg 1$ limit, which can be directly used to probe the spectral function $\mathcal{J}[\omega]$ (i.e., response function). More specifically, noting that $\hat{V} = -\hat{B}/2 = -\hat{\xi}/2$ we have

$$\Phi_q(t_f) \simeq \int_{-\infty}^{+\infty} d\omega \mathcal{F}_{\mathcal{J}}[\omega; t_f]\mathcal{J}[\omega], \qquad (36)$$

$$\mathcal{F}_{\mathcal{J}}[\omega; t_f] = -\operatorname{Im}(F^*[\omega]\eta[\omega])/4, \qquad (37)$$

and $\mathcal{F}_{\mathcal{J}}[\omega; t_f]$ forms a comb-like structure in frequency space if we fix pulse periodicity $T = t_f/2M$ and take the asymptotic large pulse number limit, i.e.,

$$\mathcal{F}_{\mathcal{J}}[\omega; t_f] \sim -\frac{M}{\omega_0} \sum_{\ell=-\infty}^{+\infty} \mathcal{A}_\ell \delta(\omega - \ell\omega_0) \quad (M \gg 1), \qquad (38)$$

$$\omega_0 = \pi/T = 2M\pi/t_f, \qquad (39)$$

where $\mathcal{A}_\ell = (4/\ell^2) \sin(\ell\pi/2)$ are constant coefficients that depend on the control sequence and can be derived using Eqs. (34) and (35).

Thus, by making use of all three levels of our spin-1 sensor, we can engineer a time-dependent quench function $\eta(t)$ that enables the construction of a standard comb-based filter function. This in turn allows spectral reconstruction of the imaginary bath response function (i.e., the spectral function $\mathcal{J}[\omega]$) over a large frequency range.

Note that Ref. [34] developed related techniques for reconstructing spatially correlated noise and response functions using multiple qubits; we stress that these are distinct from our multilevel protocols. More specifically, the multiqubit protocols crucially require 2-qubit SWAP gates in addition to standard dynamical-decoupling-type controls; further, the spectral function $\mathcal{J}[\omega]$ is not directly accessible via those existing protocols. In contrast, the quench physics in Eq. (7) provides a tool for directly probing bath response properties (i.e., its DOS), and our quench-based protocols can be straightforwardly implemented using only local spin-echo-type, or dynamical-decoupling control sequences.

## Discussion

In this work, we have shown how intrinsic quenches arise in standard $T_2$-style noise spectroscopy experiments, and how quench-induced phase-shift effects to the sensor qubit can be utilized to estimate or reconstruct the spectral function, or more general response functions of the environment. These response properties provide an independent and complementary environmental characterization from the standard NSD, and encode useful information: in combination with standard NSD, we can use the estimated spectral function to extract effective temperature of a generic nonequilibrium bath. For environments

in thermal equilibrium, the quench-enhanced QNS based on a single probe qubit also allows one to extract environmental temperature.

Our work highlights the critical role played by the initial state in controlling the effective quench physics associated with a generic $T_2$-style QNS experiment. As such, our quench formalism greatly expands previous works that considered related examples of environment-induced phase-shift effects: by analyzing quenches that arise in the most general settings of $T_2$-based QNS protocols, we show that one can engineer a generic quench operator, or a quench with complex time dependence. These generalizations allow us to further use the QPS to probe general response functions, or reconstruct the spectral function in a generic frequency range, respectively. Qubit magnetic noise spectroscopy has been used to probe electronic correlation functions in two-dimensional systems[22,61,62]; it would be interesting to apply our technique to those systems to also probe low-frequency electronic spectral function.

Our discussion so far on effective quench physics in standard $T_2$-type QNS has focused on the common case where the environment is either a quantum Gaussian bath, or where the sensor is weakly coupled to a quantum environment. While these cases make it convenient to describe the result emergence of a quench-induced sensor qubit-phase shift, the physics we have discussed is far more general. In particular, the quench has nontrivial consequences on the sensor qubit even beyond the weak coupling or Gaussian regime. In "Methods", we briefly discuss how these effect can be directly related to nonlinear response functions and noise susceptibilities of the environment. An interesting open question is how to design sensing protocols to extract those higher-order response functions. Alternatively, quantum quenches have conventionally been used to explore correlated phenomena in many-body systems[36]; our quench approach to QNS also opens up possibilities to explore new physics in these systems. We leave these to future works.

## Methods

**Generalized quenches based on arbitrary initial bath states**. Our discussion so far has focused on incidental environment quenches occurring during a generic QNS sensing protocol; for the most part, we considered a specific scenario where before the protocol starts, the environment is in the initial state described by Eq. (3). We now show that the basic quench physics we have described (and its impact on the sensor qubit) applied to a far wider set of circumstances, where the bath starts in an arbitrary initial state $\hat{\rho}_{b,i}$. This provides an entire new modality for sensing: one could deliberately prepare the environment in an interesting target state before the start of the sensing sequence, and then use the resulting quench physics (namely the influence on the sensor qubit's phase) to probe the environment.

The simplest generalization is when the environment is initially in a thermal state corresponding to some arbitrary (bath-only) Hamiltonian $\hat{H}_{b,i}$:

$$\hat{\rho}_{b,i} = e^{-\hat{H}_{b,i}/k_B T}/Z_T. \tag{40}$$

In this case, we can directly use Eq. (7) to define our quench, and identify the quench operator $\hat{V}$ via Eq. (8). We stress that in this more general case, the initial bath Hamiltonian $\hat{H}_{b,i}$ need not have any simple relation to the qubit-conditioned bath Hamiltonians $\hat{H}_{b,\uparrow(\downarrow)}$ appearing in Eq. (1). As a result, the quench operator $\hat{V}$ will now be independent of the noise operator $\hat{\xi} \equiv \hat{H}_{b,\uparrow} - \hat{H}_{b,\downarrow}$. For systems where it is possible to initialize the environment in different initial equilibrium states, this provides a powerful new way to probe the environment: different initial states yield different quenches, and hence different QPSs via Eq. (20).

An even more general scenario is when the bath starts in an arbitrary non-thermal equilibrium initial state $\hat{\rho}_{b,i}$ that has no simple relation to a static Hamiltonian. This could be achieved in numerous ways, e.g., by explicitly driving the bath[63,64]. As we have stressed repeatedly, our general quench mechanism is ultimately controlled by the initial state $\hat{\rho}_{b,i}$ of the environment. A quench will occur as part of our $T_2$-style QNS protocol any time

$$[\hat{\rho}_{b,i}, \hat{H}_{b,eff}(t)] \neq 0, \quad 0 < t < t_f. \tag{41}$$

In cases where this state was thermal, this $\hat{\rho}_{b,i}$ could easily be related to an initial bath Hamiltonian $\hat{H}_{b,i}$, which we then used to identify the quench operator $\hat{V}$ in

Eq. (8). In contrast, for our more general case, there is no unique way to identify $\hat{H}_{b,i}$. In general, we may choose any bath Hamiltonian compatible with the initial bath state, i.e., satisfying $[\hat{H}_{b,i}, \hat{\rho}_{b,i}] = 0$. The choice of $\hat{H}_{b,i}$ would then determine $\hat{V}$. We stress that this seeming ambiguity is only a choice of bookkeeping: the actual evolution of the sensor qubit (and the QPS) is of course only determined by $\hat{\rho}_{b,i}$ (see the following subsection).

Given these caveats, we now present a simple (though non-unique) method to usefully parametrize the quench in the most general case. We define the initial bath Hamiltonian $\hat{H}'_{b,i}$ and quench operator $\hat{V}'$ as the longitudinal (maximally commuting) and transverse (minimally non-commuting) components of the effective bath Hamiltonian with respect to $\hat{\rho}_{b,i}$. We can make this prescription explicit by first diagonalizing the initial bath state as $\hat{\rho}_{b,i} = \sum_{n=0}^{N} p_n \hat{P}_n$. Here the eigenvalues $p_n$ are distinct with $p_0 = 0$, and $\hat{P}_n$ is the projector onto the eigenspace corresponding to $p_n$[65]. The initial bath Hamiltonian $\hat{H}'_{b,i}$ and the quench $\hat{V}'$ can now be defined as

$$\hat{H}'_{b,i} \equiv \sum_{n=0}^{N} \hat{P}_n(\hat{H}_{b,\uparrow} + \hat{H}_{b,\downarrow})\hat{P}_n/2, \tag{42}$$

$$\hat{V}' \equiv \sum_{m,n=0;\ m \neq n}^{N} \hat{P}_m(\hat{H}_{b,\uparrow} + \hat{H}_{b,\downarrow})\hat{P}_n/2. \tag{43}$$

For concreteness, we provide an example of this procedure for a bosonic bath that couples linearly to the sensor qubit

$$\hat{H}_{b,\downarrow} = \sum_k \Omega_k \hat{b}_k^\dagger \hat{b}_k, \tag{44}$$

$$\hat{H}_{b,\uparrow} = \hat{H}_{b,\downarrow} + \sum_k \left(g_k \hat{b}_k^\dagger + \text{H.c.}\right). \tag{45}$$

We also assume that the initial bath state is not a thermal state, but a squeezed thermal state. As a result, the initial state of the sensor and bath is given by

$$\hat{\rho}_{tot}(t = 0^-) = |\downarrow\rangle\langle\downarrow| \otimes \hat{\rho}_{b,i}, \tag{46}$$

$$\hat{\rho}_{b,i} = \hat{S}(\vec{r})\, e^{-\hat{H}_{b,\downarrow}/k_B T} \hat{S}^\dagger(\vec{r})/Z_T, \tag{47}$$

where $\hat{S}(\vec{r}) = \exp(\sum_k r_k \hat{b}_k^2/2 - \text{H.c.})$ denotes the squeezing operator, with real constants $r_k$ the corresponding mode squeezing parameters[66].

Using our above prescription, we find that initial bath Hamiltonian $\hat{H}'_{b,i}$ and the quench operator $\hat{V}'$ in Eqs. (42) and (43) are given by

$$\hat{H}'_{b,i} = \sum_k \Omega_k \left[\hat{b}_k^\dagger \hat{b}_k \cosh^2 2r_k + \frac{\sinh 4r_k}{4}\left(\hat{b}_k^{\dagger 2} + \text{H.c.}\right)\right], \tag{48}$$

$$\begin{aligned}\hat{V}' = &-\sum_k \Omega_k \hat{b}_k^\dagger \hat{b}_k \sinh^2 2r_k \\ &+ \frac{1}{4}\sum_k\left(-\Omega_k \hat{b}_k^{\dagger 2} \sinh 4r_k + 2g_k \hat{b}_k^\dagger + \text{H.c.}\right).\end{aligned} \tag{49}$$

While the quench operator $\hat{V}'$ in Eq. (49) still contains an incidental contribution which depends on qubit–bath couplings $g_k$, it also includes deliberate quenches that can be tuned via the initial squeezing parameters $r_k$. We again note that it is not the only way to introduce a quench operator $\hat{V}$ satisfying Eqs. (11) and (12). However, this convention is useful for understanding effects on qubit dynamics due to the quench.

**Qubit dynamics during a standard $T_2$-type experiment**. In this subsection, we provide a detailed derivation of Eqs. (11) and (12) in the main text, which describes qubit dynamics due to pure-dephasing baths in a general $T_2$-type (e.g., spin-echo or general dynamical-decoupling) experiment. For concreteness, we first reiterate the general setup of the $T_2$-type experiment; more detail can be found in the main text. As standard, we assume that the qubit is initialized into a pure state with no qubit-bath entanglement, so that an instantaneous $\pi/2$-pulse at the beginning of the protocol ($t = 0$) prepares the system in a product state given by

$$\hat{\rho}_{tot}(t = 0^+) = |+\rangle\langle+| \otimes \hat{\rho}_{b,i}, \tag{50}$$

$$|+\rangle \equiv (|\uparrow\rangle + |\downarrow\rangle)/\sqrt{2}. \tag{51}$$

The system then evolves under a pure-dephasing-type total Hamiltonian $\hat{H}_{tot}$ for time $t_f$, while the qubit is subject to a sequence of instantaneous control $\pi$-pulses. At each instant $t$ during the time evolution, the total qubit-bath system Hamiltonian can be rewritten as

$$\hat{H}_{tot} = |\uparrow\rangle\langle\uparrow| \otimes \hat{H}_{b,\uparrow} + |\downarrow\rangle\langle\downarrow| \otimes \hat{H}_{b,\downarrow} \tag{52}$$

$$= \hat{\sigma}_0 \otimes \hat{H}_{b,avg} + \frac{\hat{\sigma}_z}{2} \otimes \hat{\xi}, \quad 0 < t < t_f, \tag{53}$$

where we introduce

$$\hat{\sigma}_0 \equiv |{\uparrow}\rangle\langle{\uparrow}| + |{\downarrow}\rangle\langle{\downarrow}|, \quad \hat{\sigma}_z \equiv |{\uparrow}\rangle\langle{\uparrow}| - |{\downarrow}\rangle\langle{\downarrow}|, \tag{54}$$

$$\hat{H}_{\mathrm{b,avg}} \equiv \frac{1}{2}(\hat{H}_{\mathrm{b},\uparrow} + \hat{H}_{\mathrm{b},\downarrow}), \quad \hat{\xi} \equiv \hat{H}_{\mathrm{b},\uparrow} - \hat{H}_{\mathrm{b},\downarrow}. \tag{55}$$

To keep our discussion general, we will assume a generic initial bath state $\hat{\rho}_{\mathrm{b,i}}$, and a corresponding initial bath Hamiltonian $\hat{H}_{\mathrm{b,i}}$ satisfying $[\hat{H}_{\mathrm{b,i}}, \hat{\rho}_{\mathrm{b,i}}] = 0$ (see also the preceding subsection). It is thus convenient to introduce a time-dependent effective bath Hamiltonian $\hat{H}_{\mathrm{b,eff}}(t)$ as

$$\hat{H}_{\mathrm{b,eff}}(t) \equiv \begin{cases} \hat{H}_{\mathrm{b,i}}, & t \le 0, \\ \hat{H}_{\mathrm{b,avg}}, & 0 < t < t_f. \end{cases} \tag{56}$$

As shown in Eq. (41), a nontrivial quench $\hat{V}$ generally arises in this standard $T_2$-type protocol, if the initial bath state $\hat{\rho}_{\mathrm{b,i}}$ does not commute with the effective bath Hamiltonian governing subsequent bath dynamics, i.e., $[\hat{\rho}_{\mathrm{b,i}}, \hat{H}_{\mathrm{b,avg}}] \neq 0$. We can rewrite $\hat{H}_{\mathrm{b,eff}}(t)$ in terms of the initial $\hat{H}_{\mathrm{b,i}}$ and this quench operator $\hat{V}$ as (see the preceding subsection and Eq. (7) in the main text)

$$\hat{H}_{\mathrm{b,eff}}(t) = \hat{H}_{\mathrm{b,i}} + \eta(t)\hat{V}, \tag{57}$$

where the quench function $\eta(t)$ vanishes, i.e., $\eta(t) = 0$, unless $0 < t < t_f$.

Transforming to the standard toggling frame with respect to qubit-control pulses, as well as the rotating frame defined by initial bath Hamiltonian $\hat{H}_{\mathrm{b,i}}$, we obtain the rotating-frame Hamiltonian

$$\hat{H}_1(t) = \eta(t)\hat{\sigma}_0 \otimes \hat{V}(t) + F(t)\frac{\hat{\sigma}_z}{2} \otimes \hat{\xi}(t), \quad 0 < t < t_f, \tag{58}$$

where $F(t)$ denotes the usual noise filter function that encodes the timing of qubit-control $\pi$-pulses, and $\hat{V}(t)$ and $\hat{\xi}(t)$ refer to the rotating-frame bath-only operators. Thus, we can compute the qubit coherence function $\langle\hat{\sigma}_-(t_f)\rangle = \langle{\uparrow}|\mathrm{Tr}_{\mathrm{bath}}[\hat{\rho}_{\mathrm{tot}}(t_f)]|{\downarrow}\rangle$ in this toggle-rotating frame as

$$\hat{\rho}_{\mathrm{tot}}(t_f) = \mathcal{T}e^{-i\int_0^{t_f} dt' \hat{H}_1(t')}\hat{\rho}_{\mathrm{tot}}(t = 0^+)\tilde{\mathcal{T}}e^{i\int_0^{t_f} dt' \hat{H}_1(t')}, \tag{59}$$

$$\Rightarrow \frac{\langle\hat{\sigma}_-(t_f)\rangle}{\langle\hat{\sigma}_-(0^+)\rangle} = \mathrm{Tr}\left\{\mathcal{T}e^{-i\int_0^{t_f} dt' \hat{H}_\uparrow(t')}\hat{\rho}_{\mathrm{b,i}}\tilde{\mathcal{T}}e^{i\int_0^{t_f} dt' \hat{H}_\downarrow(t')}\right\}, \tag{60}$$

$$\hat{H}_{\uparrow(\downarrow)}(t) \equiv \eta(t)\hat{V}(t) \pm \frac{1}{2}F(t)\hat{\xi}(t), \tag{61}$$

where $\mathcal{T}$ and $\tilde{\mathcal{T}}$ denote time- and anti-time orderings, respectively. From Eqs. (50) and (51) we have $\langle\hat{\sigma}_-(0^+)\rangle = 1/2$, so that the qubit coherence function $\langle\hat{\sigma}_-(t_f)\rangle$ can be rewritten as

$$\langle\hat{\sigma}_-(t_f)\rangle = \frac{1}{2}\mathrm{Tr}(\hat{U}_\uparrow \hat{\rho}_{\mathrm{b,i}}\hat{U}_\downarrow^\dagger), \tag{62}$$

$$\hat{U}_{\uparrow(\downarrow)} = \mathcal{T}\exp\left\{-i\int_{-\infty}^{+\infty}\left[\eta(t')\hat{V}(t') \pm \frac{F(t')}{2}\hat{\xi}(t')\right]dt'\right\}, \tag{63}$$

i.e., we obtain Eqs. (11) and (12) in the main text. Note that we adopt the convention where the filter and quench functions vanish (i.e., $F(t) = \eta(t) = 0$) unless $0 \le t \le t_f$.

From the above derivation, it is straightforward to see that the exact qubit dynamics only depends on initial bath state $\hat{\rho}_{\mathrm{b,i}}$, and Eqs. (62) and (63) (i.e., Eqs. (11) and (12) in the main text) holds for a generic quench operator $\hat{V}$ associated with any $\hat{H}_{\mathrm{b,i}}$ satisfying $[\hat{H}_{\mathrm{b,i}}, \hat{\rho}_{\mathrm{b,i}}] = 0$. For a generic bath initial state $\hat{\rho}_{\mathrm{b,i}}$, as discussed in the preceding subsection), a useful way to resolve such ambiguity in $\hat{V}$ is to choose the particular $\hat{V}' \equiv \hat{H}_{\mathrm{b,avg}}(t = 0^+) - \hat{H}'_{\mathrm{b,i}}$ as the minimally non-commuting component of $\hat{H}_{\mathrm{b,avg}}$ with respect to $\hat{\rho}_{\mathrm{b,i}}$. While this is not the only workable choice, it highlights the fact that the quench is more directly controlled by the initial bath state rather than the physical bath Hamiltonian at the beginning of the protocol. Identifying the quench in this way also allows us to compute qubit dynamics using a systematic perturbative expansion of Eqs. (62) and (63) in terms of the quench operator $\hat{V}$.

**Derivation of Eqs. (16) and (17) on leading-order dephasing and phase-shift effects to the sensor qubit due to a quenched environment.** In Eqs. (16) and (17) in the main text, we present general formulae relating leading-order bath-induced dephasing (quench-induced phase shift) effects to the bath NSD (linear response susceptibility function). While there are multiple ways to derive this result, in this subsection we provide a detailed derivation based on the Keldysh field theory technique and cumulant expansion[29]. Our approach is directly applicable to non-Gaussian baths, and elucidates the regime where Eqs. (16) and (17) become exact.

We start with Eq. (60) (of which Eqs. (11) and (12) constitute a special case), where the sensor-qubit coherence function by the end of a $T_2$-type experiment is

given by

$$\frac{\langle\hat{\sigma}_-(t_f)\rangle}{\langle\hat{\sigma}_-(0^+)\rangle} = \mathrm{Tr}\left\{\mathcal{T}e^{-i\int_0^{t_f} dt' \hat{H}_\uparrow(t')}\hat{\rho}_{\mathrm{b,i}}\tilde{\mathcal{T}}e^{i\int_0^{t_f} dt' \hat{H}_\downarrow(t')}\right\}, \tag{64}$$

$$\hat{H}_{\uparrow(\downarrow)}(t) = \eta(t)\hat{V}(t) \pm \frac{1}{2}F(t)\hat{\xi}(t). \tag{65}$$

For convenience, we introduce the Keldysh-ordered cumulant generating function (CGF) $\chi[F(t), \eta(t); t_f]$ of the bath noise and quench operators as

$$\chi[F(t), \eta(t); t_f] \equiv \ln\left[\frac{\langle\hat{\sigma}_-(t_f)\rangle}{\langle\hat{\sigma}_-(0^+)\rangle}\right]. \tag{66}$$

See Ref. [29] for discussions on the physical implications of this function. In terms of qubit dynamics, the real and imaginary parts of $\chi[F(t), \eta(t); t_f]$ correspond to qubit dephasing and bath-induced phase-shift effects, respectively.

We can now compute qubit coherence $\langle\hat{\sigma}_-(t_f)\rangle$ by perturbatively expanding the CGF $\chi[F(t), \eta(t); t_f]$ in terms of $\hat{\xi}(t)$ and $\hat{V}(t)$. The Keldysh technique offers a systematic way to perform this expansion[29,67,68]. In the Keldysh approach, for each bath operator $\hat{A}$ there are a corresponding classical field $A_{\mathrm{cl}}(t)$ and a quantum field $A_{\mathrm{q}}(t)$. Stochastic averages between these fields can be computed in a well-defined way, with respect to the so-called Keldysh action. The very construction of the Keldysh action (see, e.g., Ref. [29]) ensures that the quantum operator expectation value in Eq. (64) can be directly related to averages that only involve $\xi_{\mathrm{cl}}$ and $V_{\mathrm{q}}$. Taking the logarithm of Eq. (64), the CGF $\chi[F(t), \eta(t); t_f]$ is in turn given by

$$\chi[F(t), \eta(t); t_f] = \sum_{\ell=1}^{\infty}\sum_{m=0}^{\infty}\frac{(-i)^{\ell+m}}{\ell!m!}\chi^{(\ell,m)}[F(t), \eta(t); t_f], \tag{67}$$

$$\chi^{(\ell,m)}[F(t), \eta(t); t_f]$$
$$= \prod_{j=1}^{\ell}\left[\int_0^{t_f} dt_j F(t_j)\right]\prod_{k=\ell+1}^{\ell+m}\left[\int_0^{t_f} dt_k \eta(t_k)\right]C^{(\ell,m)}(\vec{t}_{\ell+m}), \tag{68}$$

where we define $\vec{t}_n \equiv (t_1, \ldots, t_n)$. Here $C^{(\ell,m)}(\vec{t}_{\ell+m})$ denote Keldysh-ordered cumulants, which can be directly generated from Keldysh-ordered moments of the form $\xi_{\mathrm{cl}}(t_1)\ldots\xi_{\mathrm{cl}}(t_\ell)V_{\mathrm{q}}(t_{\ell+1})\ldots V_{\mathrm{q}}(t_{\ell+m})$ and lower-order averages. Note that the terms in Eq. (67) involving cumulants of the form $C^{(\ell,0)}(\vec{t}_\ell)$ correspond to contributions solely from noise fluctuations, which would determine qubit dynamics in the absence of quenches. In contrast, the cumulants $C^{(\ell,m)}(\vec{t}_{\ell+m})$ for $m > 0$ encode bath (linear and nonlinear) response properties ($\ell = 1$), as well as noise susceptibilities ($\ell > 1$).

Without loss of generality, we assume zero-average noise operator, i.e., $\langle\hat{\xi}\rangle = 0$. The first-order contribution to the cumulant expansion in Eq. (67) thus vanishes, and the leading-order nontrivial Keldysh-ordered cumulants can be written explicitly as

$$C^{(2,0)}(\vec{t}_2) = \overline{\xi_{\mathrm{cl}}(t_1)\xi_{\mathrm{cl}}(t_2)} = \frac{1}{2}\langle\{\hat{\xi}(t_1), \hat{\xi}(t_2)\}\rangle, \tag{69}$$

$$C^{(1,1)}(\vec{t}_2) = \overline{\xi_{\mathrm{cl}}(t_1)V_{\mathrm{q}}(t_2)} = \Theta(t_1 - t_2)\langle[\hat{\xi}(t_1), \hat{V}(t_2)]\rangle, \tag{70}$$

where the bath operator average is defined with respect to $\hat{\rho}_{\mathrm{b,i}}$ as $\langle\hat{A}\rangle \equiv \mathrm{Tr}(\hat{A}\hat{\rho}_{\mathrm{b,i}})$. These leading-order cumulants can be directly related to the bath NSD $\bar{S}[\omega]$ and response susceptibility function $G^{\mathrm{R}}_{\xi V}[\omega]$, as

$$\bar{S}[\omega] \equiv \frac{1}{2}\int_{-\infty}^{+\infty} dt e^{i\omega t}\langle\{\hat{\xi}(t), \hat{\xi}(0)\}\rangle$$
$$= \int_{-\infty}^{+\infty} dt e^{i\omega t}C^{(2,0)}(t, 0), \tag{71}$$

$$G^{\mathrm{R}}_{\xi V}[\omega] \equiv -i\int_{-\infty}^{+\infty} dt e^{i\omega t}\Theta(t)\langle[\hat{\xi}(t), \hat{V}(0)]\rangle$$
$$= -i\int_{-\infty}^{+\infty} dt e^{i\omega t}C^{(1,1)}(t, 0). \tag{72}$$

By substituting the above relations into Eqs. (66) and (67), we can rewrite the leading-order contributions in terms of bath NSD and linear response susceptibility, so that we obtain

$$\frac{\langle\hat{\sigma}_-(t_f)\rangle}{\langle\hat{\sigma}_-(0^+)\rangle} = e^{-\zeta(t_f) - i\Phi(t_f)}, \tag{73}$$

$$\zeta(t_f) \simeq \frac{1}{2}\int_{-\infty}^{+\infty} dt_1 F(t_1)\int_{-\infty}^{+\infty} dt_2 F(t_2)C^{(2,0)}(\vec{t}_2)$$
$$= \int_{-\infty}^{+\infty}\frac{d\omega}{4\pi}|F[\omega]|^2\bar{S}[\omega], \tag{74}$$

$$\Phi(t_f) \simeq -i \int_{-\infty}^{+\infty} dt_1 F(t_1) \int_{-\infty}^{+\infty} dt_2 \eta(t_2) C^{(1,1)}(\vec{t}_2) \qquad (75)$$
$$= \int_{-\infty}^{+\infty} \frac{d\omega}{2\pi} F^*[\omega]\eta[\omega]G_{\xi V}^R[\omega].$$

The above equations reproduce Eqs. (16) and (17).

The cumulant expansion in Eq. (67) terminates at the second order if $\xi_{cl}(t)$ and $V_q(t)$ are Gaussian random variables, which is satisfied by linearly coupled harmonic oscillator bath models discussed in the main text. For this case, the dephasing and phase-shift expressions in Eqs. (74) and (75) become exact. Thus, we conclude that Eqs. (16) and (17), or equivalently Eqs. (74) and (75), hold exactly for Gaussian baths, where noise and quench operators can take arbitrarily strong coupling strengths. Alternatively for more general non-Gaussian baths, Eqs. (73)–(75) describe leading-order approximations for qubit dynamics in terms of bath noise and quench operators.

It is worth unpacking Eq. (17), or equivalently Eq. (75), to provide a physical understanding of the quench-induced phase. As shown in Eq. (7), the effective quench at the start of our protocol at $t=0$ suddenly turns on a term $\eta(t)\hat{V}$ in the effective bath Hamiltonian (c.f. Eq. (7)). At the linear response level, this perturbation causes a time-dependent shift in the average of the bath operator $\hat{\xi}$ that couples to the qubit. This shift is given from linear response by $\langle \delta\hat{\xi}(t)\rangle_V = \int_{-\infty}^{+\infty} dt_2 \eta(t_2) G_{\xi V}^R(t-t_2)$. Next, this induced average value of $\hat{\xi}$ has a direct consequence on the qubit: it is equivalent to a time-dependent $z$ magnetic field on the sensor qubit. This then leads to a net phase shift given by the integral of this effective field weighted by the filter function $F(t)$: $\Phi_q(t_f) = \int_{-\infty}^{+\infty} dt_1 F(t_1)\langle \delta\hat{\xi}(t_1)\rangle_V$. Connections between sensor phase shifts and linear response were also discussed in Ref. [35]. Note that our expression for the QPS can be written as $\Phi_q(t_f) = \int_{-\infty}^{+\infty} dt_1 F(t_1) \int_{-\infty}^{+\infty} dt_2 \eta(t_2) G_{\xi V}^R(t_1-t_2)$. This is similar but not identical to Eq. (15) in Ref. [35], where the filter function $F(t_1)$ erroneously occurs at a later time than the quench function $\eta(t_2)$.

**Derivation of Eq. (20) relating the quench phase shift to the imaginary part of the spectral bath response function.** In Eq. (17) of the main text, we provide a general formula to compute the quench-induced phase shift using the Green-Kubo linear response theory, which relates the phase shift to the bath susceptibility function $G_{\xi V}^R[\omega]$. We then claim that for the specific quench function $\eta(t) = \Theta(t)\Theta(t_f-t)$ emerging in a typical $T_2$-type experiment, this phase shift can be rewritten as Eq. (20), which only involves the imaginary part of response function. In this subsection, we explicitly derive the latter equation using the general formula. Recall that for a generic Gaussian bath, the QPS is given by Eq. (17) as

$$\Phi_q(t_f) = \int_{-\infty}^{+\infty} \frac{d\omega}{2\pi} F^*[\omega]\eta[\omega]G_{\xi V}^R[\omega] \qquad (76)$$

$$= \int_{-\infty}^{+\infty} dt_1 F(t_1) \int_{-\infty}^{+\infty} dt_2 \eta(t_2) G_{\xi V}^R(t_1-t_2), \qquad (77)$$

where $G_{\xi V}^R(t) \equiv -i\Theta(t)\langle[\hat{\xi}(t),\hat{V}(0)]\rangle$ is the standard Green-Kubo linear response susceptibility function. Noting that the real and imaginary parts of the spectral response function $G_{\xi V}^R[\omega]$ are related to each other via the Kramers-Kronig relation, we can rewrite the QPS in terms of only $\mathrm{Im}\, G_{\xi V}^R[\omega]$. More specifically, given that the retarded Green's function $G_{\xi V}^R(t)$ is real by definition, the imaginary part of the response function $\mathrm{Im}\, G_{\xi V}^R[\omega]$ can be written as

$$\mathrm{Im}\, G_{\xi V}^R[\omega] = \frac{1}{2i}\int_{-\infty}^{+\infty} dt\, e^{i\omega t}\left[ G_{\xi V}^R(t) - G_{\xi V}^R(-t)\right]. \qquad (78)$$

Thus, we can rewrite the QPS as

$$\Phi_q(t_f) = \int_{-\infty}^{+\infty} dt_1 F(t_1) \int_{-\infty}^{+\infty} dt_2 \eta(t_2) G_{\xi V}^R(t_1-t_2)$$
$$= \int_0^{t_f} dt_1 F(t_1) \int_0^{t_1} dt_2 \eta(t_2)[G_{\xi V}^R(t_1-t_2) - G_{\xi V}^R(t_2-t_1)] \qquad (79)$$
$$= 2\int_{-\infty}^{+\infty} \frac{d\omega}{2\pi} \mathrm{Im}\, G_{\xi V}^R[\omega]\mathcal{F}_\Phi[F,\eta(t);t_f],$$

where we have made use of the fact that the susceptibility $G_{\xi V}^R(t) = 0$ for $t < 0$. The general weighting function $\mathcal{F}_\Phi[F,\eta(t);t_f]$ is now given by

$$\mathcal{F}_\Phi[F,\eta(t);t_f] \equiv \int_0^{t_f} dt_1 F(t_1) \int_0^{t_1} dt_2 \eta(t_2) \sin\omega(t_1-t_2). \qquad (80)$$

By substituting the specific quench function $\eta(t) = \Theta(t)\Theta(t_f-t)$ into the above equation, we can explicitly compute the integral involving $\eta(t_2)$ in Eq. (80) to obtain

$$\Phi_q(t_f) = \int_{-\infty}^{+\infty} \frac{d\omega}{\pi} \frac{\mathrm{Re}\, F[0] - \mathrm{Re}\, F[\omega]}{\omega} \mathrm{Im}\, G_{\xi V}^R[\omega], \qquad (81)$$

so that for spin-echo or dynamical-decoupling control pulses with $F[0] \equiv \int_0^{t_f} F(t)dt = 0$, we recover Eq. (20). For the specific form of the quench operator $\hat{V}(t) = \hat{\xi}(t)/2$ as given by Eq. (10), we can rewrite the above expression for the QPS $\Phi_q(t_f)$ in terms of imaginary part of the response function $\mathrm{Im}\, G_{\xi\xi}^R[\omega]$ as

$$\Phi_q(t_f) = \int_{-\infty}^{+\infty} \frac{d\omega}{2\pi} \frac{\mathrm{Re}\, F[0] - \mathrm{Re}\, F[\omega]}{\omega} \mathrm{Im}\, G_{\xi\xi}^R[\omega]. \qquad (82)$$

Alternatively, for general forms of quench function $\eta(t)$, it is still possible to represent the QPS using a frequency-space integral of imaginary part of the response function, as weighted by the control functions $F[\omega]$ and $\eta[\omega]$. However, in general the integral would be nonlocal in frequency space. From the general expression in Eq. (80), we can rewrite the general transfer function for the QPS $\mathcal{F}_\Phi[F(t),\eta(t);t_f]$ as

$$\mathcal{F}_\Phi[F(t),\eta(t);t_f] = -\mathrm{Re}\, F[\omega]\,\mathrm{Im}\, \eta[\omega]$$
$$+ \mathcal{P}\int_{-\infty}^{+\infty} \frac{d\omega_1}{\pi} \frac{\mathrm{Re}\, F[\omega_1]\,\mathrm{Re}\, \eta[\omega_1]}{\omega-\omega_1}. \qquad (83)$$

As shown in Eq. (81), the above function $\mathcal{F}_\Phi[F(t),\eta(t);t_f]$ greatly simplifies when using the specific quench function $\eta(t) = \Theta(t)\Theta(t_f-t)$.

**Generalizations beyond Gaussian bath approximation.** For the case of a general environment, the qubit dynamics is still generally described by Eqs. (11) and (12). For the general case, a powerful means of attack is provided by Keldysh field theory techniques[29,67,68]. One finds that the quench does more that just induce a phase shift $\Phi_q(t_f)$ (c.f. Eq. (17)). The quench physics can also modify the noise properties of the bath, changing the dephasing function in Eq. (13).

A general way to describe these effects is to use nonlinear response theory, something that can be effectively calculated using the Keldysh approach. For weakly coupled (or Gaussian) environments, the quench only induces a nonzero average of the noise operator $\hat{\xi}(t)$ that is linear in the quench operator $\hat{V}$. In the more general cases, this shift in mean will have higher-order terms in $\hat{V}$; in addition, the symmetrized noise correlator of $\hat{\xi}(t)$ will also be modified. Keeping terms to leading orders in $\hat{V}$, these effects can formally be written in terms of nonlinear response functions as

$$\langle \delta\hat{\xi}(t)\rangle_V = \int_{-\infty}^{+\infty} dt_1 \eta(t_1) G_{\xi V}^R(t-t_1)$$
$$+ \iint_{-\infty}^{+\infty} dt_1 dt_2 \eta(t_1)\eta(t_2) G_{\xi,VV}^R(t-t_2,t_1-t_2) \qquad (84)$$
$$+ \dots,$$

$$\langle \delta\hat{\xi}(t)\delta\hat{\xi}(t')\rangle_V = \int_{-\infty}^{+\infty} dt_1 \eta(t_1) G_{\xi\xi,V}^R(t-t_1,t'-t_1) \qquad (85)$$
$$+ \dots,$$

where the susceptibility functions $G_{\xi V}^R(t)$, $G_{\xi,VV}^R(t_1,t_2)$, and $G_{\xi\xi,V}^R(t_1,t_2)$ can be systematically computed using Keldysh field theory techniques[29,68]. Note the appearance of a new function here, $G_{\xi\xi,V}^R(t_1,t_2)$. This is often referred to as a noise susceptibility, and describes how the environmental symmetrized NSD is modified by the quench. Noise susceptibilities often reveal subtle features of a physical system, and have been studied in a variety of contexts (e.g., to uncover subtle features of coherent quantum electronic transport[69]).

The upshot is that the quench physics described here is not limited to weakly coupled or Gaussian environments. For the general case, it can be described using both linear and nonlinear response functions[29,69,70]. This also highlights another utility of our quench approach to QNS: it provides in principle access to higher-order nonlinear response properties of an unknown environment using $T_2$-type measurements. More specifically, one can design general quench control protocol (e.g., making use of the multilevel probe qubit shown in Fig. 6) to generate comb-based filter functions and reconstruct these higher-order response spectral functions of a non-Gaussian quantum environment. We leave details to future work.

## Data availability
The numerical data generated in this work is available from the corresponding author upon reasonable request.

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

## Acknowledgements

We thank M. Onizhuk, G. Galli, B. D'Anjou, X. You, P. Jerger, J. Karsch, M. Fukami, N. Delegan, and F. J. Heremans for useful discussions. This work was supported as part of the Center for Novel Pathways to Quantum Coherence in Materials, an Energy Frontier Research Center funded by the U.S. Department of Energy, Office of Science, Basic Energy Sciences. A.A.C. acknowledges support from the Simons Foundation through a Simons Investigator award (Award No. 669487, AC).

## Author contributions

Y.-X.W. and A.A.C. contributed equally in the development of idea, derivation of results, and writing of paper.

## Competing interests

The authors declare no competing interests.

## Additional information

**Peer review information** *Nature Communications* thanks Félix Beaudoin and the other anonymous reviewer(s) for their contribution to the peer review this work. Peer reviewer reports are available.

