## [Peer Review File. · Nature Communications]

Reviewers' Comments:

Reviewer #1:

Remarks to the Author:

In this manuscript, the authors consider the effect of quantum quenches in the context of quantum noise spectroscopy (QNS), which aims to characterize noise due to a bath (or environment) that couples to a qubit and causes decoherence. The authors show that when the initial state of the bath does not commute with an effective system-bath coupling Hamiltonian, the back-action of the qubit on the bath associated with the start of a QNS protocol can cause a quench of the environment. Using perturbation theory, the authors show that the most apparent manifestation of this quench is an additional phase in the qubit coherence function. The authors then show how measurements of this phase can be used to characterize the spectral function of the environment, a quantity that is distinct from the more commonly studied noise spectral density, and provides new information on properties of the bath.

The authors then introduce various specific noise characterization techniques related to this quench-induced phase. Assuming a spin-boson model, the authors introduce a parametric noise characterization method that enables to measure the temperature (at equilibrium) or effective temperature (out of equilibrium) of the bath. A similar scheme is also proposed, which enables to obtain all relevant parameters of the noise spectral density and spectral function of an environment within the spin-boson model. Finally, the authors propose a more general approach (similar to dynamical-decoupling based QNS protocols) in which, exploiting tunable quenches, the spectral function of a general bath (i.e., beyond the spin-boson model) may be reconstructed, using the NV center system as an example.

To my knowledge, the idea of exploring quenches in a QNS context is completely new, and may open the door to exciting experiments probing the spectral function of the noise, a quantity that may have been overlooked in QNS. In addition to this, the ideas explored in this manuscript shed new light on results obtained by different authors in Refs. 34, 35 in which a similar phase in the qubit evolution was predicted to arise in a "rank-1" system-bath coupling model, but where the more general connection to quenches was never made. As a consequence, I believe this work will be of interest to both theorists laying down the formal foundations of quantum noise spectroscopy, and experimentalists who may be interested in new ways to probe noise that limits the performance of quantum devices, and learn about its physics through a new lens.

The methodology and analytical approach used in this manuscript appear to be scientifically sound, and the results obtained all make sense to me, in particular regarding the noise spectroscopy techniques, which I am familiar with. However, though I have encountered the Keldysh technique used in the derivations presented in the "Methods" section during my education, I have never used it in my research. Therefore, even though I doubt this would change the conclusions, I would suggest that the paper is also reviewed by an expert of the Keldysh technique before its publication.

Regarding presentation, I found that the paper was professionally written, in a way that reflected deep, original thinking, and scientific rigour. Put simply, it was a pleasure to read.

Before formally recommending acceptance of the manuscript, however, I have a few suggested improvements and questions that may hopefully help clarify a few gray zones.

The first question I have is of a relatively general scope: the authors present an interesting, new quench-induced phase shift (QPS) which they relate to phases predicted in "biased" settings in Refs. 34, 35. They also mention that such phases could be distinguished from a trivial phase arising from an external field from the fact that the QPS depends on the initial qubit state. However, non-trivial phase shifts have also been predicted to arise under non-Gaussian noise in Ref. 14. Such a phase was subsequently measured experimentally, and used to reconstruct the bispectrum of a non-Gaussian noise source in Ref. 17. Have the authors thought of any way in which the QPS may be distinguished from this non-trivial phase arising from non-Gaussianity of the environment ?

I would like to follow up with more technical questions and clarifications related to specific points in the manuscript.

1. In Results, section I, it is mentioned that there will be a control protocol. However, at this point, the total Hamiltonian does not depend on time, and presumably does not include the control. Is H_{tot} written in the toggling frame? Is it really the full Hamiltonian of the system and bath (in which case it should include the controls applied on the system), or just the system-bath coupling Hamiltonian? Perhaps it is written in the toggling frame? In any case, it may be helpful to be more explicit about the control Hamiltonian and its relationship with H_{tot} from the beginning of the section.

2. Results, Section II. It may be helpful to be a bit more explicit about the conditions leading to stationary noise. For the benefit of the reader, it may be helpful to mention what is meant by a stationary bath state, i.e., simply an eigenstate of the bath Hamiltonian.

3. In addition, the authors use the term filter function to denote $F(t)$, which encodes the timing of qubit control π pulses, in time domain. To some readers, the term "filter function" is reserved to frequency domain, and "switching function" is used in the time domain instead. This may be more intuitive to some people since the function acting as a filter is most obvious in frequency domain, while its behavior in time domain is really more analogous to a switch. If the authors still prefer to use "filter function" even in the time domain, I would suggest adding a parenthesis that clarifies $F(t)$ may also be called "switching function".

4. Above Eq. (11), I would suggest the authors mention explicitly that this is the Green-Kubo linear response function (as they do in the Methods section around Eqs. (52-53)), and cite a standard reference on the topic, since not every reader in the quantum information community is very familiar with Green's functions (in particular, experimentalists who may be interested in trying the protocol).

5. Below Eq. (14), it is mentioned that there is a critical difference between Φ_{ext} and Φ_{q} in that only the latter is sensitive to the initial state of the qubit. However, I don't think this dependence on the initial state of the qubit is obvious, in particular in Eq. (12b) it is not clear to me which quantity depends on the initial qubit state. Is the dependency on initial qubit state determined by the quench function $\eta[\omega]$? In any case, could the authors try to clarify where the dependence of the QPS on the initial qubit state arises from in a way that is immediately obvious to the reader?

5. Above Eq. (15), the authors write that for spin-echo control pulses satisfying $F[0] = 0$, the expression for the QPS further simplifies. I am puzzled by the physical meaning of this sentence. I am most familiar with switching functions toggling between -1 and 1 as in standard QNS protocols in the instantaneous pulse limit, but less so with protocols giving a vanishing switching function. Could the authors clarify which control pulses yield such a property?

6. Below Eq. (17), it would be very interesting to clarify the relationship and/or distinction between the environment spectral function $I[\omega]$ and the "quantum spectrum" studied in Refs. [16,18,34]. Mathematically, it seems like the only notable distinction is the presence of a Heaviside function in the integrand [see Eq. (11)]. I would be interested in knowing what this distinction entails, physically.

7. Below Eq. (19), it is written that these expressions "include the case where these quantities tend to a constant at $\omega=0$ ". It is not clear to me how, since there is no constant term in Eq. (19), and both expressions vanish when ω is zero. Are the actual functions defined by parts, with one part for $\omega=0$, and another for $\omega > 0$?

8. In Results, Section IV. C, the authors discuss direct thermometry under environments that are approximately Ohmic at low frequencies. For experimentalists, it would be useful to know which physical systems correspond (at least approximately) to this idealized limit.

9. In Results, Section V. A, the authors discuss how to realize frequency-space reconstruction of

response functions in NV centers. This protocol involves π pulses applied on the qubits (thus switching between states) but also, crucially, requires periodic switching between qubit subspaces. It is clear how one can switch between states -- simply apply microwaves at the appropriate transition frequency. However, it is less clear how one can switch between *subspaces*, experimentally. Could the authors please clarify this ?

10. In Methods, Eq. (44) expresses the expectation value of σ_z at time t_f in terms of an integral from $-\infty$ to ∞ , whereas the initial integrals in the time-evolution operators were from 0 to t_f . Could the authors clarify how the dependence on t_f disappeared from the integrals. In addition, no explicit dependence on t_f subsists in the expressions in Eq. (44), which I cannot understand. Could the authors clarify where is the dependence on t_f in Eq. (44) ?

Reviewer #2:

Remarks to the Author:

The manuscript of Wang and Clerk analyzes the back-action of a sensing qubit on a surrounding bath. They make the point that the Hamiltonian seen by the bath changes abruptly as the sensing qubit is switched into the sensing state. They interpret this change as a quantum quench and study the dynamic response of the bath by many-body quantum theory. They in particular analyze a coherent phase shift imprinted on the sensing qubit that arises from the coherent bath dynamics triggered by the quench. A measurement of this phase is predicted to give access to the bath temperature. Temporal concatenation of multiple quenches is demonstrated to provide a spectroscopic tool to measure the susceptibility spectrum of the bath, reminiscent of the filter-function approach to reconstruct spectral noise densities.

The result is interesting and provides a more general framework to understand and generalize several existing observations. It also creates a valuable connection between dynamical decoupling spectroscopy, linear response theory, and quantum quenches. I am finally impressed how well the paper is written. The authors have managed to give a correct account of a sophisticated theory result without blurring the big picture by excessive derivations or excessive reference to expert concepts.

The question that I cannot fully answer from the manuscript is to what extent the work affords novel insights over prior work rather than merely a novel methodology. I would like to hear the authors' view before making a definitive decision.

That the sudden change of the qubit state can be used to exert controlled backaction on the environment has been long known, and has even been used as a tool to drive nuclear spins (Tamir et al. ... Hanson, PRL 109, 137602 (2012)). Also, the prospect of measuring the temperature of a bath has been in a way anticipated by prior work (Ref 35), where a polarized ($T=0$) bath is an explicit precondition for the coherent phase to emerge.

I also have a couple of more technical remarks.

Eq (3). The authors suggest that the two terms encode action of the bath on the qubit (mediated by σ_z) and backaction of the qubit on the bath (mediated by $H_{b,up} + H_{b,down}$). I wonder whether this is an oversimplification for the first term, as this one could also create entanglement between the qubit and the bath.

"We show explicitly in Methods how one can find ... the effective quench operator ... from the initial bath state". At this point, it has not yet been explained why the bath state rather than the sensing states of the qubit would define the quench operator.

p. 3 "we first transform to an appropriate ... " there is a double-"the" at the end of the phrase.

Eqs 12 + 14. I am a bit surprised that Φ should be a real angle, if both F and B_{ext} can be arbitrary complex functions.

p. 5 the bosonic bath example is enlightening, but it would be helpful to also mention which physical system could be modeled in this way. Same for the Ohmic environment on p. 7

Eq. 19 and following discussion. I do not see how a power law could tend to a constant (other than 0 and infinity) at $\omega=0$. "Recall that ... can be characterized by a frequency-dependent temperature" this sounds like a strong statement that is not obvious to me. A reference to a textbook or a brief explanation would be helpful.

Fig. 4c - a unit is missing

Reviewer #3:

Remarks to the Author:

The theoretical work by Wang and Clerk proposes and analyzes schemes to probe the correlation and response properties of quantum baths surrounding a solid state-based qubit, like e.g. nitrogen vacancy centers in diamond. On the conceptual level, the sensing protocol is framed as a quench problem, where the bath Hamiltonian undergoes a rapid change at the beginning of the sensing procedure. In more practical terms, it is shown that the qubit decay signal hosts information on both correlation and response properties of the bath. It can therefore be used to gain quite fine grained information on both qualitative questions, such as the presence or absence of conditions of thermodynamic equilibrium condition in the bath, but also quantitative information, such as the frequency resolved determination of the bath spectral density.

The paper is very clearly written, and the work executed at high standards. The physical arguments are transparent, and the calculations well documented. That said, let me try to assess the actual novelty of the approach. The core statement is that the temporal decay of the qubit immersed in an environment is governed by the exponential of a real (decay) and imaginary (phase shift) part, which importantly encode bath correlations and response functions, respectively. The real part and its relation to correlation functions has been studied in depth before. Regarding the imaginary part, the core statement has appeared in Ref. [34] (Paz-Silva et al., Phys. Rev. A 95, 022121 (2017)) before, for a special case. I would thus rank the present work as a high-quality generalization of the latter. On the positive side, it is certainly adding some fresh twist to [34], for example in the sense of 'beating' a there articulated 'no-go theorem' on the impossibility of reconstructing the frequency resolved bath spectral function (which is no-go only under the special case assumptions in that work). Also, the interpretation of the sensing protocol of a quantum quench developed here is physically appealing. However, after all the main merit of such an applied theory work must be its actual predictivity for experiments, and these will not operationally benefit from such picture. In this view, I see the present paper rather on the incremental side, than opening up a new direction, as one would expect for a Nature Communication.

In summary, I could see that as a Nature Communications paper only if it came together with a direct experimental demonstration of the use of these theoretical considerations. But as a standalone theory work, and taking into consideration the existing work along these lines, from my view it does not clear the bar in terms of novelty or anticipated impact.

I reiterate that this is a very solid and well documented work with original results; accordingly I have only a few remarks and questions:

1. The authors argue that their approach leverages over to the case of interacting baths. While I obviously agree on that in terms of basic formulas such as (10) and (11), when it comes to their applicability to the main claims of the paper I am not as sure as the authors try to convey in their discussion. For example, for an interacting system (as also the authors remark in the core of the paper), the response function starts to depend on temperature and all other system parameters, in a nontrivial way. In such a scenario, is the present approach still useful and as unbiased as claimed? For temperature determination, I could imagine it still works. But, more generally, I was wondering to what extent the suggested schemes actually does rely on the non-interacting nature of the bath, and if there were general arguments why its predictivity (or what of it) should generalize to the interacting case.

2. Along a similar note, the authors claim that their results allow for parameter free determination

of quantities of interest, e.g., the bath temperature via the Hahn echo. On the other hand, this result is conditioned on the ohmic nature of the bath, so this clearly requires prior information about the bath. If the goal is determining the properties of the bath without prior knowledge (the ohmic assumption is often approximately realized, but when it comes to precision spectroscopy it would be more interesting to actually assess corrections), then it becomes less clear to which extent their approach is unbiased. The authors should be more careful in discussing these points.

3. A more general literature survey should be provided regarding alternative techniques of assessing bath spectral properties.

4. I do not see how the quench picture developed here can be leveraged to explore new physics (beyond the one presented here -- bath correlation and response functions), as the authors claim in their discussion. After all this is a pretty specialized setup. Some more concrete ideas would be appreciated.

Detailed responses

Response to Reviewer #1

In this manuscript, the authors consider the effect of quantum quenches in the context of quantum noise spectroscopy (QNS), which aims to characterize noise due to a bath (or environment) that couples to a qubit and causes decoherence. The authors show that when the initial state of the bath does not commute with an effective system-bath coupling Hamiltonian, the back-action of the qubit on the bath associated with the start of a QNS protocol can cause a quench of the environment. Using perturbation theory, the authors show that the most apparent manifestation of this quench is an additional phase in the qubit coherence function. The authors then show how measurements of this phase can be used to characterize the spectral function of the environment, a quantity that is distinct from the more commonly studied noise spectral density, and provides new information on properties of the bath.

The authors then introduce various specific noise characterization techniques related to this quench-induced phase. Assuming a spin-boson model, the authors introduce a parametric noise characterization method that enables to measure the temperature (at equilibrium) or effective temperature (out of equilibrium) of the bath. A similar scheme is also proposed, which enables to obtain all relevant parameters of the noise spectral density and spectral function of an environment within the spin-boson model. Finally, the authors propose a more general approach (similar to dynamical-decoupling based QNS protocols) in which, exploiting tunable quenches, the spectral function of a general bath (i.e., beyond the spin-boson model) may be reconstructed, using the NV center system as an example.

To my knowledge, the idea of exploring quenches in a QNS context is completely new, and may open the door to exciting experiments probing the spectral function of the noise, a quantity that may have been overlooked in QNS. In addition to this, the ideas explored in this manuscript shed new light on results obtained by different authors in Refs. 34, 35 in which a similar phase in the qubit evolution was predicted to arise in a "rank-1" system-bath coupling model, but where the more general connection to quenches was never made. As a consequence, I believe this work will be of interest to both theorists laying down the formal foundations of quantum noise spectroscopy, and experimentalists who may be interested in new ways to probe noise that limits the performance of quantum devices, and learn about its physics through a new lens.

The methodology and analytical approach used in this manuscript appear to be scientifically sound, and the results obtained all make sense to me, in particular regarding the noise spectroscopy techniques, which I am familiar with. However, though I have encountered the Keldysh technique used in the derivations presented in the "Methods" section during my education, I have never used it in my research. Therefore, even though I doubt this would change the conclusions, I would suggest that the paper is also reviewed by an expert of the Keldysh technique before its publication.

Regarding presentation, I found that the paper was professionally written, in a way that reflected deep, original thinking, and scientific rigour. Put simply, it was a pleasure to read.

We thank the reviewer for the summary and appreciation of our work.

Before formally recommending acceptance of the manuscript, however, I have a few suggested improvements and questions that may hopefully help clarify a few gray zones.

In the following, we address the reviewer's questions point-by-point.

The first question I have is of a relatively general scope: the authors present an interesting, new quench-induced phase shift (QPS) which they relate to phases predicted in "biased" settings in Refs. 34, 35. They also mention that such phases could be distinguished from a trivial phase arising from an external field from the fact that the QPS depends on the initial qubit state. However, non-trivial phase shifts have also been predicted to arise under non-Gaussian noise in Ref. 14. Such a phase was subsequently measured experimentally, and used to reconstruct the bispectrum of a non-Gaussian noise source in Ref. 17. Have the authors thought of any way in which the QPS may be distinguished from this non-trivial phase arising from non-Gaussianity of the environment ?

The referee asked about possible ways to distinguish the quench-induced phase shift (QPS) from phase shifts arising from non-Gaussian environmental noise. We thank the referee for bringing this important question to our attention. We wish to stress that the non-trivial phase shifts arising from non-Gaussian effects, as studied in Refs. 14,17, are due to completely distinct physics from the quench-induced phase shift (QPS). As such, there are a variety of ways to distinguish the two experimentally:

a) Perhaps the simplest scenario is when the probe qubit and its environment have a tunable coupling λ . In this case, the QPS depends quadratically on the qubit-bath coupling $\Phi_q(t_f) \propto \lambda^2$, while the leading order non-Gaussian contribution to the qubit phase $\Phi_{nG}(t_f)$ goes as $\Phi_{nG}(t_f) \propto \lambda^3$. Measuring how e.g. the Hahn-echo phase shift changes as one increases qubit-bath coupling thus lets one unambiguously distinguish the two terms from each other. For example, one could measure the qubit phase shift corresponding to qubit-bath couplings with opposite signs $\pm\lambda$: the QPS remains unchanged but the leading-order non-Gaussian phase shift would shift sign.

b) With that being said, our manuscript is primarily focused on the weak qubit-bath coupling limit (where the qubit only perturbs the bath weakly). This is the regime that realistic quantum sensors and/or quantum noise spectroscopy experiments often operate in. In this regime, the non-trivial phase shift due to non-Gaussian noise is negligible compared to the QPS (again, because the non-Gaussian effects are higher order in terms of qubit-bath coupling).

c) In the situation where one is constrained to work with a fixed qubit-bath coupling, and if non-Gaussian effects become significant compared to the QPS, the protocol discussed in Section IIIB (depicted in Fig.1b) can also be used to distinguish the quench-induced phase shift (QPS) from nontrivial non-Gaussian phases explored in Refs. 14,17. **To make this point clear, we have added the following sentence in the manuscript (paragraph following Eq. (14), last sentence):** "Further, we note that this feature lets one distinguish the QPS $\Phi_q(t_f)$ from nontrivial qubit phase shifts due to non-Gaussianity of the noise source (the latter has been studied in e.g., Refs. [14,17])." To illustrate how this idea works in a realistic setting, **we also added a detailed example of a specific system where both the QPS and non-Gaussian effects play a role: dephasing due to photon shot noise in a driven cavity (see Supplemental Note 5).**

On a related note, we wish to point out that away from the weak qubit-bath coupling regime, the *interplay* between quench physics and non-Gaussianity of the environment can lead to novel effects, as discussed in the last Method section of the manuscript. To the best of our knowledge, these quench-induced non-Gaussian effects have not been discussed in previous works (including Refs. 14,17).

I would like to follow up with more technical questions and clarifications related to specific points in the manuscript.

1. In Results, section I, it is mentioned that there will be a control protocol. However, at this point, the total Hamiltonian does not depend on time, and presumably does not include the control. Is H_{tot} written in the toggling frame ? Is it really the full Hamiltonian of the system and bath (in which case it should include the controls applied on the system), or just the system-bath coupling Hamiltonian ? Perhaps it is written in the

toggling frame ? In any case, it may be helpful to be more explicit about the control Hamiltonian and its relationship with H_{tot} from the beginning of the section.

We thank the reviewer for raising this point. Indeed we use the toggling frame at the outset of our discussion, in which case it makes sense to talk about abrupt changes to the qubit state (i.e., upon application of the qubit control pulses in the lab frame), and we also go to the rotating frame with respect to a constant free-qubit Hamiltonian (since an overall qubit rotating frequency is irrelevant to T2-type QNS experiments). **For clarification, we have changed the sentence preceding Eq. (1) to: “Transforming to the standard toggling frame set by the choice of qubit control π -pulses (see e.g. [1]), as well as the rotating frame with respect to free qubit Hamiltonian...”**

2. Results, Section II. It may be helpful to be a bit more explicit about the conditions leading to stationary noise. For the benefit of the reader, it may be helpful to mention what is meant by a stationary bath state, i.e., simply an eigenstate of the bath Hamiltonian.

To address the reviewer’s concern, **we have modified the corresponding sentence (last sentence in the first paragraph of Section II) to** “Note that as the initial bath state is stationary in our interaction picture (i.e., $[\hat{\rho}_{b,i}, \hat{H}_{b,i}] = 0$), ...”

(Note that a stationary bath could also consist of a probabilistic mixture of different eigenstates of the bath Hamiltonian, i.e. as long as the initial bath state $\hat{\rho}_{b,i}$ commutes with the bath Hamiltonian $\hat{H}_{b,i}$ then the bath is in a stationary state.)

3. In addition, the authors use the term filter function to denote $F(t)$, which encodes the timing of qubit control π pulses, in time domain. To some readers, the term “filter function” is reserved to frequency domain, and “switching function” is used in the time domain instead. This may be more intuitive to some people since the function acting as a filter is most obvious in frequency domain, while its behavior in time domain is really more analogous to a switch. If the authors still prefer to use “filter function” even in the time domain, I would suggest adding a parenthesis that clarifies $F(t)$ may also be called “switching function”.

We have changed the relevant sentence (sentence above Eq. (8a)) to “... letting $F(t)$ denote the usual filter function (also known as the switching function in time domain) that encodes the timing of qubit control π -pulses, ...”. To keep our discussion concise, in the manuscript we use the term “filter function” interchangeably for both the time- and frequency-domain functions. To avoid confusion, we use parentheses (e.g., $F(t)$) for functions defined in the time domain, and reserve square brackets (e.g., $F[\omega]$) for their Fourier transforms in the frequency space.

4. Above Eq. (11), I would suggest the authors mention explicitly that this is the Green-Kubo linear response function (as they do in the Methods section around Eqs. (52-53)), and cite a standard reference on the topic, since not every reader in the quantum information community is very familiar with Green’s functions (in particular, experimentalists who may be interested in trying the protocol).

We have modified the sentence above Eq. (11) to: “This is described by the Green-Kubo linear response function (see, e.g., [37] for a pedagogical introduction)”, where we have added Ref. 37. [37] H. Bruus and K. Flensberg, Many-body quantum theory in condensed matter physics: an introduction (Oxford University Press, Oxford, 2004).

5. Below Eq. (14), it is mentioned that there is a critical difference between Φ_{ext} and Φ_q in that only the latter is sensitive to the initial state of the qubit. However, I don’t think this dependence on the initial state of the qubit is obvious, in particular in Eq. (12b) it is not clear to me which quantity depends on the initial qubit state. Is the dependency on initial qubit state determined by the quench function $\eta[\omega]$?

In any case, could the authors try to clarify where the dependence of the QPS on the initial qubit state arises from in a way that is immediately obvious to the reader ?

In Eq. (12b), the quench operator \hat{V} explicitly depends on the initial qubit state. To address the reviewer's concern, **we have added the following sentences below Eq. (7b):** "If the total initial state is in thermal equilibrium with respect to \hat{H}_{tot} in Eq. (1), we can again define $\hat{H}_{\text{b},i}$ using Eq. (4a) as the bath Hamiltonian contingent on the initial qubit state. The quench operator \hat{V} in this case is sensitive to the initial qubit state: if the qubit was initialized in $|\uparrow\rangle$, then we would have $\hat{H}_{\text{b},i} = \hat{H}_{\text{b},\uparrow}$ and $\hat{V} = -\hat{\xi}/2$." For clarity, **we have also changed the sentence below Eq. (14) to** "There is a critical difference between Φ_{ext} and Φ_{q} : only the latter is sensitive to the initial state of the qubit (see discussion below Eq. (7b))."

5. Above Eq. (15), the authors write that for spin-echo control pulses satisfying $F[0] = 0$, the expression for the QPS further simplifies. I am puzzled by the physical meaning of this sentence. I am most familiar with switching functions toggling between -1 and 1 as in standard QNS protocols in the instantaneous pulse limit, but less so with protocols giving a vanishing switching function. Could the authors clarify which control pulses yield such a property ?

" $F[0]$ " here refers to the zero frequency $\omega = 0$ value of filter function in the frequency domain (i.e. Fourier transform of the switching function $F(t)$). When the condition $F[0] = 0$ is satisfied, the qubit switching/filter function $F(t)$ spends equal amount of time in " $F(t) = -1$ " and " $F(t) = +1$ ". For example, both Hahn echo and CPMG control pulses would satisfy this condition. For clarification, **we have modified the sentence above Eq. (15) to** "For spin-echo control pulses satisfying $F[0] \equiv \int_0^{t_f} F(t) dt = 0$, ..."

6. Below Eq. (17), it would be very interesting to clarify the relationship and/or distinction between the environment spectral function $J[\omega]$ and the "quantum spectrum" studied in Refs. [16,18,34]. Mathematically, it seems like the only notable distinction is the presence of a Heaviside function in the integrand [see Eq. (11)]. I would be interested in knowing what this distinction entails, physically.

The environmental spectral function $J[\omega]$ is equivalent to (up to a constant factor) the asymmetric part of the (unsymmetrized) quantum noise power spectral density (PSD), or the quantum noise spectrum that are studied in Refs. [16] and [18], respectively. This can be seen mathematically, noting that the quantum noise spectrum is defined as $S[\omega] \equiv \int_{-\infty}^{+\infty} dt e^{i\omega t} \langle \hat{\xi}(t) \hat{\xi}(0) \rangle$ (see, e.g., Ref. [32]), so that the spectral function (see Eq. (17) in the manuscript) can be rewritten as $J[\omega] = (1/2\pi) \int_{-\infty}^{+\infty} dt e^{i\omega t} \langle [\hat{\xi}(t), \hat{\xi}(0)] \rangle = (1/2\pi) (S[\omega] - S[-\omega])$. The spectral function $J[\omega]$ is also equivalent to the quantum self-spectra defined in Ref. [34] (note that the quantum noise spectrum studied in Refs. [16,18] is not the same as the quantum spectra defined in Ref. [34]). **To make the relation clear, we have added following sentence in the manuscript (second sentence, below Eq. (17)):** " $J[\omega]$ also corresponds to the asymmetric part of the (unsymmetrized) quantum noise spectrum [32]." We have also added citation to a review article, Ref. [32], which includes a pedagogical introduction to the various spectral properties. [32] A. A. Clerk et al., Rev. Mod. Phys. **82**, 1155 (2010).

In the manuscript, we would prefer to use the term "spectral function," because this term is routinely used in a vast amount of literature on both dissipative quantum systems (see, e.g., U. Weiss, Quantum Dissipative Systems), as well as on many-body systems (see, e.g., H. Bruus and K. Flensberg, Many-body quantum theory in condensed matter physics: an introduction). As mentioned in the manuscript (below Eq. (17)), this term also allows a straightforward physical interpretation, as it encodes the dissipative response, or an effective density of state (DOS) function of the environment.

In the second part of the question, the reviewer asked whether the Heaviside step function in Eq. (11), which defines the Green-Kubo linear response function (i.e., retarded Green's function), $G_{\hat{V}}^R[\omega] \equiv -i \int_{-\infty}^{+\infty} dt e^{i\omega t} \Theta(t) \langle [\hat{\xi}(t), \hat{V}(0)] \rangle$, somehow makes the spectral function $J[\omega]$ distinct from the quantum noise spectrum studied in Refs. [16,18]. This is not the case: **the Heaviside function does not**

ultimately enter the expression for spectral function $J[\omega]$, as it is given by the imaginary part of $G_{\xi\xi}^R[\omega]$. A straightforward calculation then shows that $J[\omega] = (1/2\pi) \int_{-\infty}^{+\infty} dt e^{i\omega t} \langle [\dot{\xi}(t), \xi(0)] \rangle$, i.e., there is no singular behaviour in the integrand associated with a step function. As a side note, the spectral function $J[\omega]$ is also related to the difference of the standard retarded and advanced Green's functions: $J[\omega] = -(G_{\xi\xi}^R[\omega] - G_{\xi\xi}^A[\omega])/2\pi i$.

7. Below Eq. (19), it is written that these expressions "include the case where these quantities tend to a constant at omega=0". It is not clear to me how, since there is no constant term in Eq. (19), and both expressions vanish when omega is zero. Are the actual functions defined by parts, with one part for omega=0, and another for omega > 0 ?

We thank the reviewer for pointing out this typo. **We have made the following change to the manuscript (sentence below Eq. (20b), which corresponds to Eq. (19b) in the old manuscript):** "... includes the case where these quantities tend to a constant **asymptotically as $\omega \rightarrow 0^+$** ."

8. In Results, Section IV. C, the authors discuss direct thermometry under environments that are approximately Ohmic at low frequencies. For experimentalists, it would be useful to know which physical systems correspond (at least approximately) to this idealized limit.

We thank the reviewer for this suggestion, and **we have added the following sentence in the manuscript (first paragraph of Section IVD, corresponding to Section IVC of the old manuscript):** "Baths with an Ohmic spectral density $J[\omega]$ are both extremely well studied theoretically, and are good descriptions of various dissipative environments [53]. Perhaps the best known examples are the voltage and current fluctuations (i.e., Johnson-Nyquist noise [54,55]) of an electromagnetic environment described by an impedance that is frequency-independent at low frequencies. Such electromagnetic environments are relevant to many systems, including superconducting qubits [56-59]."

We have also added Refs. 53-59:

[53] A. O. Caldeira and A. J. Leggett, Quantum tunneling in a dissipative system, *Ann. Phys.* **149**, 374 (1987).

[54] J. B. Johnson, Thermal agitation of electricity in conductors, *Phys. Rev.* **32**, 97 (1928).

[55] H. Nyquist, Thermal agitation of electric charge in conductors, *Phys. Rev.* **32**, 110 (1928).

[56] M. H. Devoret, Quantum fluctuations in electrical circuits, in *Quantum Fluctuations*, edited by S. Reynaud, E. Giacobino, and J. Zinn-Justin (Elsevier, Amsterdam, 1997), Chap. 10, pp. 351–386.

[57] B. Peropadre, D. Zueco, D. Porras, and J. J. García-Ripoll, Nonequilibrium and nonperturbative dynamics of ultrastrong coupling in open lines, *Phys. Rev. Lett.* **111**, 243602 (2013).

[58] F. Yan et al., The flux qubit revisited to enhance coherence and reproducibility, *Nat. Commun.* **7**, 12964 (2016).

[59] P. Forn-Díaz et al., Ultrastrong coupling of a single artificial atom to an electromagnetic continuum in the nonperturbative regime, *Nat. Phys.* **13**, 39 (2017).

*9. In Results, Section V. A, the authors discuss how to realize frequency-space reconstruction of response functions in NV centers. Their protocol involves pi pulses applied on the qubits (thus switching between states) but also, crucially, requires periodic switching between qubit subspaces. It is clear how one can switch between states -- simply apply microwaves at the appropriate transition frequency. However, it is less clear how one can switch between *subspaces*, experimentally. Could the authors please clarify this ?*

Experimentally, switching between the "subspaces" requires the ability to independently addressing the $|0\rangle \leftrightarrow |-1\rangle$ and the $|0\rangle \leftrightarrow |+1\rangle$ transitions (or more generally, at least two different transitions between the three states of the probe). For NV centers, this can be realized by applying an external magnetic field (to differentiate the transition frequencies), and applying microwaves at the appropriate transition frequencies.

For example, if one wants to switch from using $\{|\uparrow\rangle = |-1\rangle, |\downarrow\rangle = |0\rangle\}$ to using $\{|\uparrow\rangle = |0\rangle, |\downarrow\rangle = |+1\rangle\}$ as the qubit subspace, this can be achieved by subsequently applying two π -pulses between the $|0\rangle \leftrightarrow |+1\rangle$ and the $|0\rangle \leftrightarrow |-1\rangle$ transitions. Given that these transitions are of distinct frequencies, this can be implemented experimentally by subsequently applying two microwave pulses at the corresponding transition frequencies.

10. In Methods, Eq. (44) expresses the expectation value of σ_x at time t_f in terms of an integral from $-\infty$ to ∞ , whereas the initial integrals in the time-evolution operators were from 0 to t_f . Could the authors clarify how the dependence on t_f disappeared from the integrals. In addition, no explicit dependence on t_f subsists in the expressions in Eq. (44), which I cannot understand. Could the authors clarify where is the dependence on t_f in Eq. (44) ?

The dependence on t_f in the integral in Eq. (46b) (i.e., Eq.(44b) in the old manuscript) was implicitly implemented, because (without loss of generality) we define the time-domain filter and quench functions to be zero unless $0 \leq t \leq t_f$. To address the reviewer's concern, **we have added the following sentence below Eq. (46b):** "Note that we adopt the convention where the filter and quench functions vanish (i.e., $F(t) = \eta(t) = 0$) unless $0 \leq t \leq t_f$."

Response to Reviewer #2

The manuscript of Wang and Clerk analyzes the back-action of a sensing qubit on a surrounding bath. They make the point that the Hamiltonian seen by the bath changes abruptly as the sensing qubit is switched into the sensing state. They interpret this change as a quantum quench and study the dynamic response of the bath by many-body quantum theory. They in particular analyze a coherent phase shift imprinted on the sensing qubit that arises from the coherent bath dynamics triggered by the quench. A measurement of this phase is predicted to give access to the bath temperature. Temporal concatenation of multiple quenches is demonstrated to provide a spectroscopic tool to measure the susceptibility spectrum of the bath, reminiscent of the filter-function approach to reconstruct spectral noise densities.

The result is interesting and provides a more general framework to understand and generalize several existing observations. It also creates a valuable connection between dynamical decoupling spectroscopy, linear response theory, and quantum quenches. I am finally impressed how well the paper is written. The authors have managed to give a correct account of a sophisticated theory result without blurring the big picture by excessive derivations or excessive reference to expert concepts.

We appreciate the reviewer's encouraging remarks on our work.

The question that I cannot fully answer from the manuscript is to what extent the work affords novel insights over prior work rather than merely a novel methodology. I would like to hear the authors' view before making a definitive decision.

That the sudden change of the qubit state can be used to exert controlled backaction on the environment has been long known, and has even been used as a tool to drive nuclear spins (Tamiriau ... Hanson, PRL 109, 137602 (2012)). Also, the prospect of measuring the temperature of a bath has been in a way anticipated by prior work (Ref 35), where a polarized ($T=0$) bath is an explicit precondition for the coherent phase to emerge.

We thank the reviewer for raising interesting and relevant questions about connection to previous works, and we address each in turn below.

i) In the first part of the question, the reviewer asked about the relation between our work and previous work(s) that used the qubit as a tool to drive its surrounding nuclear spins. The reviewer is certainly right

that the work by Taminiau et al. (Taminiau ... Hanson, PRL 109, 137602 (2012)) made use of sudden changes of the qubit state to apply controlled backaction on the environment. However, there is a key difference from our approach: Taminiau et al.'s approach is explicitly based on having a detailed model of a rather simple quantum environment (a collection of nuclear spins that do not interact with each other), and then using this knowledge to control the nuclear spins via the qubit state. In contrast, our approach is for a setting where almost nothing is known about the environment, and where “quench” physics is a way of learning more about the bath that would be possible using only standard dephasing measurements. The goal of our approach is to learn something about an unknown environment. **The validity of our general results does not depend on microscopic details of the environment – the environment could be bosonic, fermionic, spin, or a mixture, interacting or non-interacting.**

There are other stark differences. The Taminiau et al. work considers a non-interacting nuclear spin bath prepared in a completely unpolarized (infinite temperature) initial state. For such a bath, the quench-induced phase shift (QPS), which is the main focus of our work, would be strictly zero. Mathematically, one can see this from the Green-Kubo formula, which reduces to the trace of a commutator and hence vanishes if the bath is finite dimensional and in a maximally mixed state. The lack of a QPS is also reflected by the fact that their exact solution for NV center spin dynamics does not include any non-trivial phase shift contributions (i.e. Eq. (2) in the Taminiau et al. paper is always real), including the weak qubit-bath coupling limit where our results would be applicable. In this sense, the work by Taminiau et al. is based on completely distinct back-action physics from the QPS discussed in the manuscript.

ii) In the second half of the question, the reviewer pointed out that previous work (Ref. 35, Kwiatkowski et al., Phys. Rev. B 101, 155412 (2020)) anticipated the possibility of temperature measurement using a related phase shift generated by polarized spin baths and a “biased” qubit-bath coupling. However, our work shows that related “quench phase shifts” can arise in a far more general setting, without the need of any special form of coupling. We believe this realization (and the realization that the QPS can be directly controlled by the initial qubit state) represent significant advances over previous work, and will make these ideas applicable to a far greater range of systems. **For clarification, we have rewritten the corresponding sentences in the manuscript (p. 2, left column, second paragraph):** “We note that related phase shifts were discussed in previous works as an anomalous effect emerging in T_2 -type QNS protocols in systems with an unusual ‘biased’ qubit-environment coupling [34,35]. In contrast, as we show the quench-induced phase shifts can in fact arise in a far wider set of systems, including ones with an ‘unbiased’ coupling that according to previous works, would exhibit no extra phase shift.” We also note that Ref. 35 was explicitly about a specific kind of microscopic bath. Our work is in contrast extremely general. To highlight this generality, and the fact that one does not require a special biased coupling, **we have added discussion of two realistic examples of microscopic quantum baths:** $1/f$ noise (p. 7, Section IVC) and photon shot noise in a driven cavity (Supplementary Note 5). Both these examples go beyond the “biased” qubit-bath coupling framework in Ref. 35 and related previous works. We numerically computed the Hahn-echo signal due to the non-trivial QPS in both cases.

More generally, we wish to stress that bath temperature estimation is only one aspect of the various potential applications of our work, and hence is not the only respect in which our work is an advance over previous studies. For instance, in Section IVA, we provided a concrete protocol that could probe spectral functions of nonequilibrium baths in general. We also discussed a general protocol making use of a probe qubit based on multilevel systems to reconstruct the environmental spectral function (Section V), which utilizes quench physics that is entirely distinct from the biased coupling framework in Ref. 35. Thus, we believe our work is more than a novel methodology when compared to Ref. 35.

As a side note, Ref. 35 only commented that the *bias-induced* phase shift “disappears when the state of the environment is maximally mixed,” so that “... to observe this shift, the inverse temperature β has to be finite, in the case of thermal equilibrium.” Apart from these general comments, the main part of quantitative predictions in Ref. 35 are based on an extremely specific type of non-interacting spin bath model. Simply stating that a non-trivial phase shift requires a non-infinite bath temperature does not

necessarily indicate that the phase shift can be used to measure temperature: in general, a non-zero phase shift may also depend on various bath parameters apart from the temperature. In this sense, we think the concrete temperature estimation protocol in our work also constitutes a new result that would be of interest to the quantum sensing community.

I also have a couple of more technical remarks.

Eq (3). The authors suggest that the two terms encode action of the bath on the qubit (mediated by σ_z) and backaction of the qubit on the bath (mediated by $H_{b,up} + H_{b,down}$). I wonder whether this is an oversimplification for the first term, as this one could also create entanglement between the qubit and the bath.

We apologize for any confusion. There is actually no contradiction here: the corresponding sentence in the manuscript was focused on time evolution of the qubit (i.e., looking at the reduced density matrix of the qubit), in which case the generation of **qubit-bath entanglement manifests itself as apparent dephasing of the qubit**. More specifically, the effect from the bath on the qubit can be encapsulated by the qubit-bath interaction $\hat{\sigma}_z \otimes \hat{\xi}$: the averaged value of the bath noise operator $\hat{\xi}$ determines bath-induced frequency shift on the qubit, whereas fluctuations in the bath operator $\hat{\xi}$ will cause qubit dephasing.

For clarification, we have implemented following changes in the manuscript (sentences below Eq. (3)): “This suggests a simple picture for **understanding the qubit** evolution during the protocol: the qubit dephases due to coupling to the bath noise operator $\hat{\xi}$, while the bath evolves under an effective averaged bath Hamiltonian. **Note that these two processes are not independent, as the effective bath Hamiltonian would affect $\hat{\xi}$ during time evolution and hence influence qubit dynamics.**”

“We show explicitly in Methods how one can find ... the effective quench operator ... from the initial bath state”. At this point, it has not yet been explained why the bath state rather than the sensing states of the qubit would define the quench operator.

We thank the reviewer for this observation. For clarification, **we have changed the sentence below Eq. (7b) to:** “If the total initial state is in thermal equilibrium with respect to \hat{H}_{tot} in Eq. (1), we can again define $\hat{H}_{b,i}$ using Eq. (4a) as the bath Hamiltonian contingent on the initial qubit state. The quench operator \hat{V} in this case is sensitive to the initial qubit state: if the qubit was initialized in $|\uparrow\rangle$, then we would have $\hat{H}_{b,i} = \hat{H}_{b,\uparrow}$ and $\hat{V} = -\hat{\xi}/2$. If the total system is initially out of equilibrium, the definitions in Eqs. (5) and (6) can describe quench physics corresponding to a much wider range of nonequilibrium initial bath states, even beyond the specific case in Eqs. (2b) and (7b). In the more general case, $\hat{H}_{b,i}$ in Eq. (5) is directly controlled by the initial bath state (see Methods).”

p. 3 “we first transform to an appropriate ... ” there is a double-“the” at the end of the phrase.

We thank the reviewer for pointing out this typo. Upon suggestion from reviewer #1, **we have moved the second half of the sentence to the beginning of Section I, and the corrected sentence now reads:** “Transforming to the standard toggling frame set by the choice of qubit control π -pulses (see e.g. [1]), as well as the rotating frame with respect to free qubit Hamiltonian...” **We have also changed the original sentence to** (p. 3, first paragraph of Section II, second sentence): “We first transform to an appropriate ..., and **we again assume the standard toggling frame defined by qubit control pulses.**”

Eqs 12 + 14. I am a bit surprised that Phi should be a real angle, if both F and B_ext can be arbitrary complex functions.

$\Phi_q(t_f)$ in Eq. (12b) and $\Phi_{ext}(t_f)$ in Eq. (14) are guaranteed to be real, because by definition, the frequency-space functions $F[\omega]$, $\eta[\omega]$, $G_{\xi V}^R[\omega]$, and $B_{ext}[\omega]$ entering these equations are **Fourier**

transforms of real-valued functions in the time domain (and are not arbitrary complex functions).

As a result, the real (imaginary) parts of the integrands in both equations are guaranteed to be even (odd) functions in frequency ω , so that when integrated over the entire frequency space, the imaginary part of the integral is zero. This would also become more transparent if the equations are rewritten in the time domain (see, e.g., Eq. (55) on p.15, corresponding to Eq. (53) in the old manuscript, where $F(t_1)$, $\eta(t_2)$, and $G_{\xi V}^R(t_1 - t_2)$ are purely real by definition); in the manuscript we mostly write the integral in Fourier domain to explain the sensing protocol.

p. 5 the bosonic bath example is enlightening, but it would be helpful to also mention which physical system could be modeled in this way. Same for the Ohmic environment on p. 7

We thank the reviewer for this suggestion. We have implemented corresponding changes in the manuscript:

p. 5, the paragraph on bosonic bath example (left column, last sentence):

“This bosonic bath model can be used to describe a variety of phononic or electromagnetic dephasing environments [38,39].”

We have also added citations to Refs. [38,39], which discussed the bosonic bath models in more detail: [38] U. Weiss, Quantum Dissipative Systems (World Scientific, 2012).

[39] A. J. Leggett et al., Dynamics of the dissipative two-state system, Rev. Mod. Phys. **59**, 1 (1987).

p. 7, on Ohmic environment (Section IVD, first paragraph):

“Baths with an Ohmic spectral density $\mathcal{J}[\omega]$ are both extremely well studied theoretically, and are good descriptions of various dissipative environments [53]. Perhaps the best known examples are the voltage and current fluctuations (i.e., Johnson-Nyquist noise [54,55]) of an electromagnetic environment described by an impedance that is frequency-independent at low frequencies. Such electromagnetic environments are relevant to many systems, including superconducting qubits [56-59].”

We have also added Refs. 53-59:

[53] A. O. Caldeira and A. J. Leggett, Quantum tunneling in a dissipative system, Ann. Phys. **149**, 374 (1987).

[54] J. B. Johnson, Thermal agitation of electricity in conductors, Phys. Rev. **32**, 97 (1928).

[55] H. Nyquist, Thermal agitation of electric charge in conductors, Phys. Rev. **32**, 110 (1928).

[56] M. H. Devoret, Quantum fluctuations in electrical circuits, in Quantum Fluctuations, edited by S. Reynaud, E. Giacobino, and J. Zinn-Justin (Elsevier, Amsterdam, 1997), Chap. 10, pp. 351–386.

[57] B. Peropadre, D. Zueco, D. Porras, and J. J. García-Ripoll, Nonequilibrium and nonperturbative dynamics of ultrastrong coupling in open lines, Phys. Rev. Lett. **111**, 243602 (2013).

[58] F. Yan et al., The flux qubit revisited to enhance coherence and reproducibility, Nat. Commun. **7**, 12964 (2016).

[59] P. Forn-Díaz et al., Ultrastrong coupling of a single artificial atom to an electromagnetic continuum in the nonperturbative regime, Nat. Phys. **13**, 39 (2017).

Eq. 19 and following discussion. I do not see how a power law could tend to a constant (other than 0 and infinity) at $\omega=0$. "Recall that ... can be characterized by a frequency-dependent temperature" this sounds like a strong statement that is not obvious to me. A reference to a textbook or a brief explanation would be helpful.

1. For the sentence following Eq. (20) (i.e., Eq. (19) in the old manuscript): we thank the reviewer for pointing out this typo. **We have corrected the corresponding sentence, which now reads (sentence below Eq. (20b)):** “Note this includes the case where these quantities tend to a constant **asymptotically as $\omega \rightarrow 0^+$** .”

2. Regarding the sentence on frequency-dependent temperature: we draw the reviewer’s attention to discussion on p. 5 (right column, paragraph containing Eq. (19)) and references cited therein

(Refs. 32,43,44, or Refs. 32,40,41 in the old manuscript). For nonequilibrium baths and for a given bath operator, we can use the corresponding noise spectral density $\bar{S}[\omega]$ and spectral function $J[\omega]$ to **define** a frequency-dependent temperature $T_{\text{eff}}[\omega]$, so that the fluctuation-dissipation relation ($\bar{S}[\omega] = \pi J[\omega] \coth \omega/2k_B T_{\text{eff}}[\omega]$) is satisfied at every frequency ω . As we discussed in the manuscript, this concept is useful because a non-trivial frequency dependence in the resulting effective temperature $T_{\text{eff}}[\omega]$ would indicate that the bath is initially in non-equilibrium. To address the reviewer's concern, **we have added a numbered equation, Eq. (19) on p. 5, where we first introduce $T_{\text{eff}}[\omega]$ and provide the relevant references, and we refer to it again in later discussion (paragraph below Eq. (20b))**: “As shown in Eq. (19), a general, non-thermal environment can always be characterized by a frequency-dependent effective temperature $T_{\text{eff}}[\omega]$.”

Fig. 4c - a unit is missing

We thank the reviewer for this observation. In the caption of Fig. 5 (previously Fig. 4), we have added (second to last sentence): “**We assume $\alpha = 1$ in all panels.**”

Response to Reviewer #3

The theoretical work by Wang and Clerk proposes and analyzes schemes to probe the correlation and response properties of quantum baths surrounding a solid state-based qubit, like e.g. nitrogen vacancy centers in diamond. On the conceptual level, the sensing protocol is framed as a quench problem, where the bath Hamiltonian undergoes a rapid change at the beginning of the sensing procedure. In more practical terms, it is shown that the qubit decay signal hosts information on both correlation and response properties of the bath. It can therefore be used to gain quite fine grained information on both qualitative questions, such as the presence or absence of conditions of thermodynamic equilibrium condition in the bath, but also quantitative information, such as the frequency resolved determination of the bath spectral density.

The paper is very clearly written, and the work executed at high standards. The physical arguments are transparent, and the calculations well documented. That said, let me try to assess the actual novelty of the approach. The core statement is that the temporal decay of the qubit immersed in an environment is governed by the exponential of a real (decay) and imaginary (phase shift) part, which importantly encode bath correlations and response functions, respectively. The real part and its relation to correlation functions has been studied in depth before. Regarding the imaginary part, the core statement has appeared in Ref. [34] (Paz-Silva et al., Phys. Rev. A 95, 022121 (2017)) before, for a special case. I would thus rank the present work as a high-quality generalization of the latter. On the positive side, it is certainly adding some fresh twist to [34], for example in the sense of 'beating' a there articulated 'no-go theorem' on the impossibility of reconstructing the frequency resolved bath spectral function (which is no-go only under the special case assumptions in that work). Also, the interpretation of the sensing protocol of a quantum quench developed here is physically appealing. However, after all the main merit of such an applied theory work must be its actual predictivity for experiments, and these will not operationally benefit from such picture. In this view, I see the present paper rather on the incremental side, than opening up a new direction, as one would expect for a Nature Communication.

We appreciate the reviewer's positive comments, and their thoughtful remarks on the relation of our work to Ref. [34]. However, we respectfully but strongly disagree with the reviewer's conclusions that our work is merely a “**high-quality generalization of**” Ref. [34], or that actual experiments “**will not operationally benefit**” from our work in terms of predictive power. We outline our reasons for this below.

i) The reviewer stated that “Regarding the imaginary part, the core statement has appeared in Ref. [34] (Paz-Silva et al., Phys. Rev. A 95, 022121 (2017)) before, for a special case.” We think this is untrue:

While Ref. [34] discussed a related phase shift effect, Paz-Silva et al. attributed the phase shift to a very specific form of qubit-bath coupling Hamiltonian (e.g. “M2 models” defined in Ref. [34]), and explicitly stated that similar phase shift must vanish in systems with a different form of qubit-bath coupling (i.e., “M1” models defined in Ref. [34], see comments on p. 7 below Eq. (30) in Ref. [34]). In contrast, we presented a new quench-based approach to understand the emergence of the quench-induced phase shift (QPS). This different way to understand the physics is **more than a matter of physical interpretation, because in contrary to the viewpoint taken in previous work, the quench physics is relevant to a much wider range of systems than previously realized. For instance, we showed that the quench-induced phase shift can arise in systems where Ref. [34] would predict to exhibit zero extra phase shift** (i.e., “M1” models coined in Ref. [34]). As such, our approach makes it easier to use the quench physics in experiments for quantum noise spectroscopy applications, by designing experimental protocols whose applicability does not rely on the nature of the qubit-bath coupling. For these reasons, we think it is incorrect to state that “the core statement” of the manuscript “has appeared in Ref. [34] ... for a special case.”

ii) The reviewer also questioned how, compared to previous works, our work provides new predictive power that would benefit experiments. As we stressed in the manuscript, our work shows how the quench physics opens up new ways to using quantum noise spectroscopy to probe unknown quantum environments. This includes measuring the QPS using standard T_2 -type experimental setups to independently extract the bath spectral function. As discussed in Section IV, having access to this information could help answer the question of whether the environment is in thermal equilibrium, and if so, what is the bath temperature. These will be relevant questions for people interested in understanding dephasing noise sources of practical qubits, as well as general quantum sensing applications. Further, as the reviewer pointed out, we discussed a general protocol making use of a probe qubit based on multilevel systems to reconstruct the environmental spectral function (Section V), utilizing quench physics that is entirely distinct from the framework in Ref. [34] and related works. **None of these results are in any sense alluded to in Ref. [34].** Specifically on using the phase shift for the purpose of quantum noise spectroscopy, the main relevant equation in Ref. [34] is Eq. (30), which just related the phase shift to the “quantum self-spectrum,” and nowhere in the work is it implied that the phase shift could be useful for single-qubit QNS.

For clarification, **we have rewritten the corresponding sentences in the manuscript (p. 2, second paragraph on the left column):** “We note that related phase shifts were discussed in previous works as an anomalous effect emerging in T_2 -type QNS protocols in systems with an unusual ‘biased’ qubit-environment coupling [34,35]. In contrast, as we show the quench-induced phase shifts can in fact arise in a far wider set of systems, including ones with an ‘unbiased’ coupling that according to previous works, would exhibit no extra phase shift.” Further, **we have added detailed numerical studies of QPS in two realistic quantum bath models, based on 1/f noise in superconducting qubits (p. 7, Section IVC) and photon shot noise in a driven cavity (Supplementary Note 5),** respectively. Both examples go beyond the framework used in Ref. [34], in that one would **not** expect a non-trivial phase shift (encoding the bath response) to arise, if one were to use the “M1/M2” classification scheme in Ref. [34] (and related previous works) that is solely based on the form of qubit-bath coupling operators.

In summary, I could see that as a Nature Communications paper only if it came together with a direct experimental demonstration of the use of these theoretical considerations. But as a standalone theory work, and taking into consideration the existing work along these lines, from my view it does not clear the bar in terms of novelty or anticipated impact.

We sincerely hope that the reviewer will reconsider their decision. As we pointed out in the detailed response above, the quench approach opens up new ways to using quantum noise spectroscopy (QNS) to probe the spectral function of generic, unknown quantum environments. Towards this general goal, we discussed a variety of quench-enhanced QNS protocols in the manuscript. None of these novel protocols are suggested in previous works that touched on a related phase shift effect, nor are they simple

modifications of ideas presented in those works. To further address the reviewer's concern about relation of our work to previous works, we have added two numerical studies of the quench phase shift, based on realistic models of experimentally relevant quantum environments (see the detailed response above); both examples clearly go beyond existing works (including Ref. [34] that discussed a related phase shift effect). We hope our detailed responses and the new additions make it clear how our work is an advance over previous studies, letting it meet the high bar for novelty and significance required for Nature Communications.

I reiterate that this is a very solid and well documented work with original results; accordingly I have only a few remarks and questions:

1. The authors argue that their approach leverages over to the case of interacting baths. While I obviously agree on that in terms of basic formulas such as (10) and (11), when it comes to their applicability to the main claims of the paper I am not as sure as the authors try to convey in their discussion. For example, for an interacting system (as also the authors remark in the core of the paper), the response function starts to depend on temperature and all other system parameters, in a nontrivial way. In such a scenario, is the present approach still useful and as unbiased as claimed? For temperature determination, I could imagine it still works. But, more generally, I was wondering to what extent the suggested schemes actually does rely on the non-interacting nature of the bath, and if there were general arguments why its predictivity (or what of it) should generalize to the interacting case.

We thank the reviewer for raising relevant questions about practical applicability of our approach, and we address each part of the question in turn below.

a) The reviewer asked whether our approach is still applicable to interacting systems, where the response function might exhibit non-trivial temperature dependence. The answer is definitely **yes**: in the weakly-coupled bath regime we are generally interested in, the **temperature estimation protocols do not rely on the response function being independent of temperature, nor on how the spectral function depends on any other system parameters**. We apologize if discussions on the bosonic bath example in the old manuscript might have caused any confusion, and **we have revised the corresponding sentences to (p. 5 right column, first paragraph): "In this simple bosonic case, $\mathcal{J}[\omega]$ is completely independent of the environmental state. However, we stress that our results in subsequent sections remain valid for interacting baths, and/or baths that are not purely bosonic."**

b) The reviewer questioned if the quench approach is "still useful and as unbiased as claimed" in generally interacting systems, and/or when the response function has non-trivial dependence on temperature or other system parameters. The answer is again yes. As we stressed in multiple places in the manuscript, the quench approach lets one (independently) extract the bath spectral function in a wide range of systems. It is well understood in a variety of fields that the spectral function provides a very useful characterization, even (and especially) for strongly interacting systems. There have been a large body of research dedicated to measuring and calculating spectral functions in those systems, ranging from high- T_c superconductors (W. Rantner and X.-G. Wen, Phys. Rev. Lett. 86, 3871 (2001)), spin liquids (L. Savary and L. Balents, Rep. Prog. Phys. 80 016502 (2017)), to magnons (A. L. Chernyshev and M. E. Zhitomirsky, Phys. Rev. B 79, 144416 (2009)), etc.; understanding e.g. how the spectral function depends on specific parameters may reveal valuable information for understanding physics of the system, and our approach offers one way to achieve that.

c) The reviewer also asked for general arguments why our approach generalizes to interacting baths. We reiterate that our approach is valid in the weak qubit-bath coupling limit. In the more general case with stronger qubit-bath coupling, one would also expect nonlinear and non-Gaussian effects to arise. In the manuscript, we also showed how these nonlinear, non-Gaussian effects can be described (see the last Method section). To the best of our knowledge, the new effects due to the interplay between quench and non-Gaussianity of the environment have not been discussed in previous works.

In terms of experiments, one could always try to work in the weakly qubit-environment coupling limit so that nonlinear effects are negligible in the measurements. We remark that this (verifying whether the measurement results can be described by linear response) is a general problem for any spectroscopy tools attempting to probe linear response functions, and we do not think it is a drawback due to our specific approach.

2. Along a similar note, the authors claim that their results allow for parameter free determination of quantities of interest, e.g., the bath temperature via the Hahn echo. On the other hand, this result is conditioned on the ohmic nature of the bath, so this clearly requires prior information about the bath. If the goal is determining the properties of the bath without prior knowledge (the ohmic assumption is often approximately realized, but when it comes to precision spectroscopy it would be more interesting to actually assess corrections), then it becomes less clear to which extent their approach is unbiased. The authors should be more careful in discussing these points.

There seems to be some confusion here: our temperature estimation protocol in no way requires the environment to have an Ohmic spectral density. This was stated explicitly in Section IVA of the manuscript. The referee's confusion likely resulted from the fact that later in the manuscript (Sec. IVD, or IVC in the old manuscript), we do present more specialized results for the specific case of an Ohmic bath. We believe this specific case study is still useful, as Ohmic baths arise in a number of physical settings, and are also extremely well studied in the dissipative quantum systems literature. **To avoid any confusion, we have changed the title of Section IVA** to "General estimation protocol for sub-, super-, and Ohmic environments." We have also added a sentence (p. 2, first paragraph): "As we discuss, such information lets us determine the temperature of a thermal equilibrium environment, making only mild assumptions encompassing a wide range of realistic environments (including sub-, super-, and Ohmic environments, environments generating $1/f$ noise, etc.)."

We agree with the reviewer that one should be careful with discussing prerequisites for determining properties of the bath, and we discussed these issues in detail in the manuscript. More specifically for temperature estimation, we made mild assumptions about the bath spectral function $\mathcal{J}[\omega]$: i) the spectral function depends on frequency as a power-law in the low-frequency limit (i.e., $\mathcal{J}[\omega] \propto \omega^s$ as $\omega \rightarrow 0^+$, see Eq. (20b)), and ii) the spectral function does not exhibit too strong divergence at low frequencies, which is satisfied by most physical environments (see p. 6, right column, first paragraph). We are happy to discuss further if the reviewer still has concerns about assumptions required for the temperature estimation protocols.

The reviewer also asked about corrections to our results when e.g., the Ohmic spectral function is only approximately valid. We draw the reviewer's attention to the last paragraph on p. 8, where we explored influence of a finite-frequency cutoff on the quench phase shift dynamics. As we showed, such cutoffs on the Ohmic behavior in the spectral function lead to a finite crossover protocol time: above the crossover time, our asymptotic results on the quench phase shift become valid.

3. A more general literature survey should be provided regarding alternative techniques of assessing bath spectral properties.

Regarding alternative techniques for assessing bath spectral properties, the old manuscript cited several works on T2-type (see, e.g., Refs. 3,5 for theoretical, and Refs. 7,24 for experimental studies), as well as relaxometry-type noise spectroscopy (see, e.g., Ref. 31 for theoretical, and Refs. 16,18 for experimental studies). We also discussed the comparison between the two techniques in Supplementary Note 1. We diligently searched in the literature, and have added Refs. S5,S6 (see Supplementary Note 1) that demonstrated different techniques to characterize the noise spectrum of the same system (surface magnetic noise in diamond, and charge noise in quantum dots, respectively). If the reviewer has additional relevant references in mind, we would greatly appreciate them explicitly listing these for us.

[S5] T. Roskopf et al., Investigation of surface magnetic noise by shallow spins in diamond, Phys. Rev. Lett. **112**, 147602 (2014).

[S6] E. J. Connors, JJ Nelson, and J. M. Nichol, Charge-noise spectroscopy of Si/SiGe quantum dots via dynamically-decoupled exchange oscillations, arXiv:2103.02448.

4. I do not see how the quench picture developed here can be leveraged to explore new physics (beyond the one presented here -- bath correlation and response functions), as the authors claim in their discussion. After all this is a pretty specialized setup. Some more concrete ideas would be appreciated.

The reviewer asked the question of how the quench approach can be used to explore new physics “beyond ... bath correlation and response functions.” We would like to note that the ability to measure correlation functions (including noise spectral densities and spectral functions) of unknown systems is a general pathway to explore new physics, which has already been demonstrated in a variety of systems, ranging from biological systems to condensed matter systems including, e.g., strongly interacting Fermi gases. A host of measurement techniques have been solely dedicated to this goal (not to mention the huge variety of theoretical methods that have been developed to calculate these functions both analytically and numerically). Given the ubiquity and interest in spectral functions, we do believe our work is extremely general, and could provide a route to uncovering new physics (e.g., as discussed in the paper, determining whether a given environment is in thermal equilibrium or not).

The reviewer also raised concerns about the quench approach being a “specialized setup.” We respectfully but strongly disagree that our work is based on a specialized setup: it uses a generic dephasing interaction between a qubit and an environment, which has been demonstrated in almost all types of quantum systems. As we stressed in the manuscript, the quench approach allows one to directly probe the spectral function of an unknown environment, giving rise to a new sensing modality that is applicable in various quantum sensing platforms (including, e.g., quantum dots qubits, superconducting qubits, and qubits based on solid-state defects).

The reviewer finally requested more concrete examples. To illustrate how the quench approach can be useful for probing a generic environment, we added two extensive examples based on realistic quantum bath models, describing $1/f$ charge noise in superconducting qubits (Section IVC) and photon shot noise (Supplemental Note 1), respectively. For each of those examples, we also discussed how measuring the quench-induced phase shift could shed new light on physics of the corresponding quantum environment.

Additional notes:

While revising the manuscript, we noticed following typos that we have amended:

1. In Eq. (4a), the right hand side should read “ $\text{Tr}_{\text{qb}}[\hat{\rho}_{\text{qb}}(t)\hat{H}_{\text{tot}}(t)]$ ”.

2. In the old manuscript, the caption for Fig. 4 (and corresponding main text discussion) missed a constant factor of 2 in the equation describing linewidth of the Lorentzian peak (in the spectral function). We have corrected the relevant equation in the caption to “ $2\Gamma_{\text{min}}/\omega_c = 0.2 < 1$ ”, and the main text discussion to “... Lorentzian peak with linewidth $2\Gamma_{\text{min}} = 2\epsilon\omega_c = 0.2\omega_c$ ” (p. 8, last paragraph).

Reviewers' Comments:

Reviewer #1:

Remarks to the Author:

The authors have meticulously addressed all the concerns that I presented in my first report, and improved their manuscript accordingly. I now unambiguously recommend this manuscript for publication in Nature Communications.

Reviewer #2:

Remarks to the Author:

I recommend the revised manuscript for publication in Nature Communications. The only critical question among all three referees seems to be the impact of the manuscript. Thinking twice, I believe that, while some aspects may have appeared in prior work, the new language and concept introduced by the authors will simplify the discussion so substantially that the paper is likely to become a milestone reference for future work. This in my opinion is sufficient to justify publication in Nat Comm.

The technical quality of the manuscript is outstanding, as it has been from the very start.

Reviewer #3:

Remarks to the Author:

The authors managed to significantly sharpen the novel points of their work compared to the previously existing literature: They convinced me that the approach is more predictive than that of Ref. [34], by carefully working out specific examples in the new version of the manuscript. In addition, they argue convincingly that the case of interacting baths is covered as well, so that indeed no prior assumptions on the bath have to be made. They clarified the manuscript in this direction.

Based on these improvements, I can recommend this work for publication in Nature Communications.

We thank all reviewers for their time and efforts in reviewing our work, and for providing useful feedback that helped us improve the manuscript.

Detailed responses:

Response to Reviewer #1

The authors have meticulously addressed all the concerns that I presented in my first report, and improved their manuscript accordingly. I now unambiguously recommend this manuscript for publication in Nature Communications.

We again thank the reviewer for their constructive feedback, which helped us improve the manuscript. We appreciate the reviewer's endorsement of our manuscript.

Response to Reviewer #2

I recommend the revised manuscript for publication in Nature Communications. The only critical question among all three referees seems to be the impact of the manuscript. Thinking twice, I believe that, while some aspects may have appeared in prior work, the new language and concept introduced by the authors will simplify the discussion so substantially that the paper is likely to become a milestone reference for future work. This in my opinion is sufficient to justify publication in Nat Comm. The technical quality of the manuscript is outstanding, as it has been from the very start.

We are very grateful to the reviewer for their encouraging remarks, and we are happy to hear that the reviewer now supports publication of our work in Nat. Comm.

Response to Reviewer #3

The authors managed to significantly sharpen the novel points of their work compared to the previously existing literature: They convinced me that the approach is more predictive than that of Ref. [34], by carefully working out specific examples in the new version of the manuscript. In addition, they argue convincingly that the case of interacting baths is covered as well, so that indeed no prior assumptions on the bath have to be made. They clarified the manuscript in this direction. Based on these improvements, I can recommend this work for publication in Nature Communications.

We are glad to hear that our response and the added examples have successfully convinced the reviewer of the novelty of our work. It was also gratifying to read that the reviewer now seems to agree with us about the applicability of our approach to interacting baths. Again, we appreciate the reviewer for bringing up relevant questions, which helped us improve our manuscript. We thank the reviewer for their recommendation of our work for publication in Nature Communications.